# Parameterizations for global thundercloud corona discharge distributions

Sergio Soler[1], Francisco J. Gordillo-Vázquez[1], Francisco J. Pérez-Invernón[1], Patrick Jöckel[2], Torsten Neubert[3], Olivier Chanrion[3], Victor Reglero[4], and Nikolai Østgaard[5]

[1]Instituto de Astrofísica de Andalucía (IAA), CSIC, PO Box 3004, 18080 Granada, Spain
[2]Deutsches Zentrum für Luft- und Raumfahrt, Institut für Physik der Atmosphäre, Oberpfaffenhofen, Germany
[3]National Space Institute, Technical University of Denmark (DTU Space), Kongens Lyngby, Denmark
[4]Image Processing Laboratory, University of Valencia, Valencia, Spain
[5]Birkeland Centre for Space Science, Department of Physics and Technology, University of Bergen, Bergen, Norway

**Correspondence:** F. J. Gordillo-Vázquez (vazquez@iaa.es)

**Abstract.** Four parameterizations, distinguishing between land and ocean, have been developed to simulate global distributions of thundercloud streamer corona discharges (also known as Blue LUminous Events or BLUEs) mainly producing bluish optical emissions associated to the second positive system of $N_2$ accompanied by no (or hardly detectable) 777.4 nm light emission. BLUEs occur globally about 12 times less frequently (Soler et al., 2022) than lightning flashes. The four schemes are based on nonlinear functions of the cloud top height (CTH), the product of the convective available potential energy (CAPE) and total precipitation (TP), the product of CAPE and specific cloud liquid water content (CLWC), and the product of CAPE and specific cloud snow water content (CSWC). Considering that thunderstorms occur on hourly timescales, these parameterizations have been tested using hourly ERA5 data (except for CTH, not available in ERA5) for the meteorological variables considered, finding that the proposed BLUE schemes work fine and are consistent with observations by ASIM. Moreover, the parameterizations have been implemented in a global chemistry-climate model that generates annual and seasonal global distributions for present day and end of 21st century climate scenarios. Present day predictions are in reasonable agreement with recent observations by the Atmosphere Space Interaction Monitor (ASIM). Predictions for the end of the 21st century suggest BLUE occurrence rates that range between 13 % larger ($\sim$ 3 % per K), and 52 % larger ($\sim$ 13 % per K) than present day average occurrence of BLUEs.

## 1 Introduction

The availability of regular space observations of total (intra cloud and cloud to ground) lightning since 1995 has generated large datasets that have allowed to derive annual and seasonal geographical distributions of total lightning, resulting in an annual average flash rate of $\sim$ 45 $\pm$ 2 flashes s$^{-1}$ between $\pm$ 52° latitude (Christian et al., 2003; Cecil et al., 2014; Blakeslee et al., 2020). Prediction of global total lightning flash rate and geographical distribution are increasingly important, since lightning is a frequent natural hazard, considered a proxy for severe weather, a cause of large wildfires (Komarek, 1964; Pyne et al., 1998; Latham and Williams, 2001; Pérez-Invernón et al., 2021, 2022, 2023), and a direct source of nitric oxide (NO) (Huntrieser et al.,

2002; Schumann and Huntrieser, 2007; Pérez-Invernón et al., 2022) in the troposphere that impacts the balance of important upper troposphere lower stratosphere (UTLS) chemical species such as nitrogen dioxide (NO$_2$), ozone (O$_3$) and key oxydizing radicals such as hydroxyl (OH) and hydroperoxyl (HO$_2$), (Schumann and Huntrieser, 2007; Finney et al., 2016; Gordillo-Vázquez et al., 2019). Besides this, recent studies also suggest a direct production of OH and HO$_2$ by lightning strokes (Brune et al., 2021). All these reasons supported the need to incorporate lightning into chemistry-climate models.

The sub-grid spatial dimensions of lightning require their parameterization using different input meteorological variables and functional forms (Price and Rind, 1992; Grewe et al., 2001; Allen and Pickering, 2002; Finney et al., 2014; Luhar et al., 2021). The implementation of lightning parameterizations in different global chemistry-climate models (Tost et al., 2007; Romps et al., 2014a; Finney et al., 2014; Gordillo-Vázquez et al., 2019) have been tested against total lightning observations from low earth orbit (Christian et al., 2003; Cecil et al., 2014; Blakeslee et al., 2020) and, very recently, also using data from geostationary satellites (Zhang et al., 2021).

Corona discharges, occurring both in the lab and in thunderclouds, are characterized by cold ionization waves known as streamers. Corona discharges are formed by numerous streamers. The electromagnetic counterpart of thundercloud corona discharges are Narrow Bipolar Events (NBEs) (Rison et al., 2016; Soler et al., 2020). They produce bluish optical emissions (250-450 nm), leading to the adoption of the term Blue Luminous Events (BLUEs) for their optical counterpart.

While the hot and thermal air plasma in lightning stroke channels mostly excites atomic species like oxygen atoms released from thermal dissociation of O$_2$ leading to 777.4 nm optical emissions typical of lightning flashes, streamer corona discharges are cold non-thermal plasmas where only heavy particles are cold and electrons are very hot (up to 10 eV). Thus corona discharges are able to activate (excite) molecular species like N$_2$, O$_2$ and H$_2$O by non-thermal equilibrium electron-impact collisions (Gordillo-Vázquez and Pérez-Invernón, 2021), which cause distinct bluish optical emissions mostly associated to second positive system of N$_2$ radiative de-excitations.

Research results since the early 1970s indicate that, in addition to lightning, thundercloud leaderless kilometer scale corona electrical discharges formed by hundreds of millions of streamers (Liu et al., 2019; Cooray et al., 2020; Li et al., 2021; Soler et al., 2022) are relatively common, $\sim 45 \pm 2$ lightning flashes s$^{-1}$ vs 3.5 Blue flashes s$^{-1}$, that is, $\sim 12$ times less frequent than global average number of lightning flashes in thunderclouds around the globe. In particular, recent laboratory experiments (Jenkins et al., 2021) suggest that observations during thunderstorms reporting sudden local enhancements of O$_3$ (Shlanta and Moore, 1972; Brandvold et al., 1996; Zahn et al., 2002; Minschwaner et al., 2008; Brune et al., 2021), and OH and HO$_2$ (Brune et al., 2021) could be associated to dim leaderless corona discharges (BLUEs) in storm clouds (Brune et al., 2021). These episodes suggest a probable regional atmospheric chemistry impact of thundercloud coronas, a subject which is still poorly quantified (Gordillo-Vázquez and Pérez-Invernón, 2021).

In this study we present four parameterizations, distinguishing between land and ocean, to simulate global distributions of thundercloud corona discharges producing BLUEs (Soler et al., 2020, 2021, 2022; Li et al., 2021). The proposed storm cloud corona schemes are based on a non-linear dependence of cloud top height (CTH), and on non-linear combinations of pairs of meteorological parameters such as convective available potential energy (CAPE) and total precipitation (TP), CAPE and cloud liquid water content (CLWC), or CAPE and cloud snow water content (CSWC), which are all available from satellite

data and atmospheric reanalysis (used to build the parameterizations) and in global chemistry-climate models. Some of these meteorological variables (CTH, Price and Rind (1992), TP, Allen and Pickering (2002); Romps et al. (2014b, 2018), CAPE, Romps et al. (2014b, 2018)) have been previously used to build different lightning parameterizations (Price and Rind, 1992; Allen and Pickering, 2002; Romps et al., 2014b, 2018). Other variables, like CSWC and CLWC, had not been used before (to the best of our knowledge). Both CSWC and CLWC can contribute to the electrification of the thundercloud since collision of graupel and ice water crystals at temperatures less than 253 K results in a negative charge transfer to the graupel that falls to lower regions of the cloud. The lighter, positive charged ice crystals stay in the higher regions of the cloud.

The physics behind CAPE (used in three of the four developed corona discharge schemes) relies on findings shown in (1) Soler et al. (2021), and (2) Husbjerg et al. (2022). Figure S12 by Soler et al. (2021) first showed that the seasonal CAPE is, in general, stronger in regions with more BLUEs. On the other hand, Figure 5b by Husbjerg et al. (2022) showed that the CAPE associated to thunderstorms producing lightning flashes have a median value of 1000 J/Kg while thunderstorms producing BLUEs require stronger convection than needed for lightning alone. The CAPE found in the scenarios of thunderstorms that produce BLUEs range (median values) between 1280 J/Kg (slow BLUES, that is, those buried in the thunderclouds) and 1570 J/Kg (fast BLUES, that is, those appearing in the top of thunderclouds). As indicated by Husbjerg et al. (2022), a CAPE larger than 2000 J/Kg usually indicates deep convection. Cells generating fast BLUE occur in 25% of the time in the region of deep convection. For cells generating only slow BLUE discharges it is 17% while for regular lightning, only 10% of the events have a CAPE greater than 2000 J/Kg. Therefore, there is a strong link between deep convection and the generation of BLUE discharge events. Another consequence is that, in general, the occurrence of lightning is more probable since they do not require so much the presence of deep convection. These results lead us to consider CAPE as a plausible meteorological variable to track the occurrence of BLUEs.

Regarding the terms *fast* and *slow* mentioned above, please note that both of them underlie the scattering of the light emitted by BLUEs in thunderclouds. Since *fast* BLUEs are located in the cloud top, the scattering of their light emission is smaller (than that of *slow* BLUEs located in the bottom of the cloud). Consequently, the rise and decay times of the light curves (as seen by ASIM photometers) are faster than the rise/decay times of the light curve associated with *slow* BLUEs that last longer.

Most previous lightning parameterizations have been tested in a number of global atmosphere circulation models. This was done to explore how the different lightning schemes compare with available lightning observations in the present, to establish correlations with meteorological / climatic patterns and to predict possible future lightning occurrence global rates and geographical distributions in the context of a variety of future climatic scenarios. Our goal here is to procede similarly using BLUE parameterizations since models allow looking into the future (end of the 21st century) to reach preliminary answers to how BLUEs geographical distribution and global occurrence rate will change in a warmer atmosphere. Therefore, the parametrizations are first tested on reanalysis data. After that, a global chemistry-climate model is used to (i) test the corona schemes against present day climatic scenarios (both annual and seasonal), when observations are available by the Atmosphere Space Interaction Monitor (ASIM), and to (ii) approximately predict the occurrence rate and annual geographical distribution of thundercloud coronas in future (2091-2095) climate scenarios.

The next section describes the data, observations and modeling employed to build and test the proposed thundercloud corona parameterizations. Section 3 explains the procedures followed to develop the schemes for cloud corona discharges. Section 4 evaluates the climatological performance of the storm cloud corona parameterizations at present day and for an end of the 21st century climate scenario. The last section of the paper presents the main conclusions.

## 2 Data description, observations and modeling

### 2.1 ECMWF ERA5 and COPERNICUS CLARA datasets

The European Centre for Medium-Range Weather Forecasting (ECMWF) provides the ERA5 global atmospheric reanalysis data product (Hersbach et al., 2020). The ERA5 data cover the Earth on a spatial resolution of $0.25°$ latitude and longitude and resolve the atmosphere using 137 levels from the surface up to a pressure of 0.01 hPa ($\sim$ 80 km height). Single or surface level data are also available. ERA5 combines large amounts of historical observations into global estimates using advanced modelling and data assimilation systems. ERA5 provides hourly (also sub-daily and monthly) estimates of a large number of atmospheric, land and oceanic climate variables.

In order to build our thundercloud corona parameterizations we have selected as input variables annual averages of the cloud top height (CTH), convective available potencial energy (CAPE) and total precipitation (TP), which are single (surface) level variables, and the annual averages of the specific cloud liquid water content (CLWC) and specific snow liquid water content (CSWC) at 450 hPa. CAPE, TP, CLWC and CSWC are taken from ERA5 (Hersbach, H. et al., 2018a, b; Hersbach et al., 2020). CTH (not available in ERA5) is taken from the CLARA product family (Karlsson et al., 2017) of the Essential Climate Variable (ECV) Cloud Properties of COPERNICUS (the European Union's Earth observation programme).

Note that hourly data have been averaged to obtain daily values and then averaged again into monthly values and yearly values. We have tested the BLUE parameterizations with hourly data (except for CTH for which only monthly data are available) and the result is shown in Figure 1 (see section 3) based on hourly data of the meteorological parameters used.

### 2.2 Observations used

We use global observations of nighttime thundercloud corona discharges (also known as Blue LUminous Events or BLUEs) recorded by the high sampling rate (100 kHz) photometer in the near UV (337 nm/4 nm) of the Modular Multispectral Imaging Array (MMIA) that is part of ASIM (Chanrion et al., 2019; Soler et al., 2020, 2021). ASIM is aboard the International Space Station (ISS) and, due to the inclination ($\sim 52°$) of the ISS orbit, locations near the equator are observed less frequently than those at higher latitudes.

The worldwide corona observations used here have been recently published (Soler et al., 2022) and span a period of two years of MMIA level 1 (calibrated) data from 1 April 2019 to 31 March 2021. In particular, we use the annual and seasonal averages associated with the global distribution of BLUEs obtained by the algorithm described in Soler et al. (2021) adding an

extra step for filtering high energy (and cosmic ray) candidates. The resultant distribution (used here) is named GD-2 (Soler et al., 2022).

MMIA observations of storm cloud coronas exhibit strong features in the 337 nm/4 nm photometer with negligible (barely above the noise level 0.4 $\mu$W/m$^2$) signal in the 777.4 nm/5 nm photometer, which is also continuously monitored (Soler et al., 2020, 2021; Li et al., 2021).

## 2.3 Modeling

As an illustration of their applicability, the developed BLUE parameterizations have been incorporated into a chemistry-climate model. We use the ECHAM / MESSy Atmospheric Chemistry (EMAC) model, which is a chemistry-climate model that couples the fifth generation European Center HAMburg general circulation model (ECHAM5) and the second version of Modular Earth Submodel System (MESSy) to link multi-institutional computer codes, known as MESSy submodels (Jöckel et al., 2010, 2016). Such submodels are used to describe tropospheric and middle atmosphere processes and their interaction with oceans, land, and influences coming from anthropogenic emissions.

The thundercloud corona parameterizations described below are used in MESSy for usage within the ECHAM / MESSy Atmospheric Chemistry (EMAC) model. Cloud corona schemes are implemented as a new element of EMAC to account for atmospheric electricity phenomena that, so far, only includes lightning parameterizations. The present day simulations without interactive chemistry are performed without lightning chemical emissions. For the climate simulations, for the end of the 21st century, we have used the lightning parameterization by Grewe et al. (2001) based on the updraft flux of mass. The variables CAPE, convective precipitation, and specific cloud liquid water content (CLWC) are calculated by the submodel CONVECT (Tiedtke, 1989; Nordeng, 1994), while the large scale precipitation is imported from the CLOUD submodel (Roeckner et al., 2006). The total precipitation is calculated as the sum of the convective and the large scale precipitation. Following the same approach as the LNOX submodel for the calculation of lightning (Tost et al., 2007), corona frequencies are ignored below a certain cut-off, in order to avoid introducing artifacts in the simulations. We set the corona frequency to zero if the cloud thickness is lower than 3 km.

Two sets of simulations are performed, one that covers the present day climatic state and another one for the end of the 21st century under the Representative Concentration Pathway 6.0 (RCP6.0). RCP6.0 simulations are used to estimate future occurrence and geographical patterns of thundercloud coronas. We do not have enough computational resources to run all the possible future scenarios. Therefore, we had to choose only one of them. We chose the RCP6.0 scenario that is one of the two intermediate stabilization pathways (higher medium). The EMAC simulations are performed with 720 seconds time step length and in the T42L90MA resolution, i.e., with a 2.8° × 2.8° quadratic Gaussian grid in latitude and longitude with 90 vertical levels starting at the surface and reaching up to the 0.01 hPa pressure level (Jöckel et al., 2016). Present day simulations are set-up by using the namelist setup for purely dynamical simulations (referred to as the E5 setup, no chemistry) in the mode of free-running simulations. Present-day simulations are started in January, 2000 using ERA-Interim reanalysis meteorological fields (ECMWF, 2011) as initial conditions. Please note that ERA-Interim starting field has no impact on the simulation (since we are adopting ERA5 for the meteorological parameters). The RCP6.0 simulation is set-up following the

155 simulation RC2-base-04 of Jöckel et al. (2016) and Pérez-Invernón et al. (2023). The sea surface temperatures (SSTs) and the sea-ice concentrations (SICs) are prescribed from simulations with the Hadley Center Global Environment Model version 2 - Earth System (HadGEM2-ES) Model (Collins et al., 2011; Bellouin et al., 2011). Projected mixing ratios of the greenhouse gases and SF6 are incorporated from Eyring et al. (2013). Anthrophogenic emissions are taken from monthly values provided by Fujino et al. (2006) for the RCP6.0 scenario. We refer to Jöckel et al. (2016) for more details about the simulation set-up.

We start the 6-years RCP6.0 in January, 2090 and consider 1 year of spin-up to reach equilibrium. Section 4 below shows a comparison of the results of implementing thunderstorm corona schemes in EMAC for the years 2091 to 2095 with present day observations (recorded by ASIM) and with simulations for the years 2000 to 2009. We are aware of the limitations of this comparison in terms of the short period of available space observations. However, it is important to highlight that this is the first global scale continuous observations of BLUEs and, though limited, we consider it is worth showing such comparison between

present day and the warmer atmosphere expected for the end of the 21st century. The present day simulations cover 10 years. The projection (for the end of the 21st century) simulations span 5 years. It is important to note that both simulations cover more than 2 years, which are the total number of years used to develop the parameterizations. We do not simulate more years because considerable computational resources are needed. The projection simulations are initialized by using the prescribed conditions of year 2090, previously obtained from the transient climate simulations RC2-base-04 by Jöckel et al. (2016). With

this approach, the climate state of 2090 (as projected for the RCP6.0 scenario) is already established and the production of BLUEs, since based on the meteorological parameters, adapts quasi immediately. Finally, please note that despite the projection simulation spans only 5 years instead of 10 years, the mean and the standard deviation of the obtained global rate of BLUEs for the end of the 21st century are significantly larger than those obtained from the present day simulations.

## 3  Thundercloud Corona Parameterizations

Electrical activity in thunderclouds in the form of lightning flashes have been previously correlated with CAPE (Williams et al., 1992; Pawar et al., 2012), with precipitation (Battan, 1965; Petersen and Rutledge, 1998; Allen and Pickering, 2002), and even with a linear combination of both of them (Romps et al., 2014a, 2018).

The experimental finding that thunderstorm electrification (in terms of substantial charge transfer) needs the presence of water droplets (Saunders et al., 1991) indicates that the liquid water content in thunderclouds can be a good proxy for electrical

activity.

We have chosen the convective available potential energy (CAPE) as a proxy of deep convection as shown by Ukkonen and Mäkelä (2019). Also, as illustrated in Figure S12 of the supplementary material of Soler et al. (2021), the seasonal CAPE shows that, in general, there are stronger CAPEs in regions with more BLUEs. However, CAPE can also be very high in the ocean where it is not that common to find many BLUEs so that it is important to distinguish between land and ocean. Accompanying

Figure S12 in Soler et al. (2021), the seasonal CAPE vs BLUEs per second relationships were quantified using the Pearson linear correlation coefficient (R) (varying between -1 and +1, with +1 being perfect linear correlation, 0 null linear correlation,

and -1 perfect linear anti-correlation). The correlations resulted better for the zonal distributions ($0.77 < R < 0.89$, with DJF and MAM being the best) than for the meridional distributions ($0.42 < R < 0.64$, with DJF and JJA being the best).

Following the above results, Figure 5b in Husbjerg et al. (2022), showed that by clustering the BLUE discharge data set, cells which generate *fast* (close to cloud top) BLUE discharges have a median CAPE of 1390 $\mathrm{Jkg}^{-1}$ compared to 1128 $\mathrm{Jkg}^{-1}$ for cells generating only *slow* (deep in the cloud) BLUE discharges, further indicating that stronger cells are more likely to generate fast BLUE discharge. For comparison, the median CAPE for lightning flashes was 816 $\mathrm{Jkg}^{-1}$ in Husbjerg et al. (2022).

The above results lead us to use CAPE as a plausible meteorological variable to track the occurrence of BLUEs. Thus, three of the proposed parameterizations are based on CAPE times another meteorological variable like the total precipitation (previously used in lightning schemes like the one proposed in Romps et al. (2014b) or the Cloud Liquid (or Snow) Water Contents (CLWC or CSWC), which presence in thunderclouds can contribute to cloud electrification. He et al. (2022) developed a parameterization based on the product of CAPE times the charging rate of collisions between graupel and other types of hydrometeors. Finally, we used the Cloud Top Height (CTH) variable to somehow test its "quality" as it was proposed in the popular lightning scheme presented by Price and Rind (1992).

We propose four parameterizations for corona discharges in thunderclouds based on (i) a nonlinear dependence of the cloud top height (CTH (km)) and nonlinear combinations of (ii) the convective available potential energy (CAPE (J /$\mathrm{Kg}_{air}$)) and the total precipitation (TP (m/day), (iii) CAPE and the specific cloud liquid water content (CLWC ($\mathrm{Kg}_{liquid}$ / $\mathrm{Kg}_{moist-air}$)), and on (iv) CAPE and the specific cloud snow water content (CSWC ($\mathrm{Kg}_{ice}$ / $\mathrm{Kg}_{moist-air}$)), where the CLWC and CSWC are defined as the mass of cloud liquid and snow water droplets per kilogram of the total mass of moist air.

We build the thundercloud corona schemes by using them in a two year data period (1 April 2019 to 31 March 2021). We consider the annual average number of thundercloud coronas per second given in each grid cell by the GD-2 distribution provided by Soler et al. (2022), and the annual average of the meteorological variables. The CTH, CAPE, TP and the specific CLWC and CSWC are taken within $2° \times 2°$ grid cells where cloud coronas take place according to GD-2.

The detailed procedure to build the corona parameterizations follows a number of steps: First, we take both $2° \times 2°$ maps − observed global annual averaged cloud corona occurrence rate (BLUEs $\mathrm{s}^{-1}$) and annually averaged of monthly averaged meteorological variables chosen as proxys − and transform them into 1D arrays. Second, we sort the meteorological dataset array following ascending values. Third, we sort the coronas array doing the same index changes we made in the second step. Fourth, we divide each array into 40 chunks, cutting each interval where the percentile distribution is a divisor of 2.5 % (neglecting the geographical position). We choose 40 chunks as an optimum, if we take more chunks we may have too many repeated values between one chunk and the next, and if we take less values it may not be enough to characterize the curve. Fifth, we calculate the mean of each chunk, this will result in 40 points for the predictor (meteorological data) and 40 points for the predictand (corona data). Note that the previous procedure is performed at global scale. Sixth, we split the grid into land and ocean cells, for that purpose we create a mask of land and ocean cells, then split each $2° \times 2°$ cell into 400 cells of $0.1° \times 0.1°$ size, and calculate in the center of that cell if it is over land (1) or over ocean (0). To finish, we assign to each $2° \times 2°$ cell the average of the 400 cells of $0.1° \times 0.1°$ (the average is always in the intervall [0,1]). Then, if the value is larger than 0.5 it is land, otherwise it is ocean. Finally, we approximate, separating between land (l) and ocean (o), the data

with the equation $y^{l,o} = \alpha^{l,o} \times x^{\beta^{l,o}}$ by a least squares fit, where $x$ are the values of the selected meteorological variables used as proxys and $y^{l,o}$ corresponds to the occurrence rate (in events s$^{-1}$) of thundercloud coronas over land ($y^l$) and ocean ($y^o$), respectively. Note that the chosen mathematical formulae ($y^{l,o} = \alpha^{l,o} \times x^{\beta^{l,o}}$) for the parameterization of BLUEs are formally similar to previous ones for lightning flashes (Price and Rind, 1992; Michalon et al., 1999; Luhar et al., 2021; Romps et al., 2014b) including the product of two magnitudes, a constant $\alpha$ multiplied by a variable $x$ raised to a power ($\beta$), where $x$ can be a single meteorological variable (Price and Rind, 1992; Michalon et al., 1999; Luhar et al., 2021) or the product of two meteorological variables (Romps et al., 2014b). However, as indicated in the introduction, the physics behind the corona schemes is not exactly the same as that underlying lightning parameterizations.

Figure 1(a)-(d) represents the global annual average (based on hourly data for CAPE, TP, CLWC and CSWC, and monthly data for CTH) of the nighttime corona occurrence rate (coronas s$^{-1}$) versus CTH, CAPE $\times$ TP, CAPE $\times$ CLWC and CAPE $\times$ CSWC, which result in the four proposed schemes $C_{F1}^{l,o} = \alpha_1^{l,o} \times (\text{CTH})^{\beta_1^{l,o}}$, $C_{F2}^{l,o} = \alpha_2^{l,o} \times (\text{CAPE} \times \text{TP})^{\beta_2^{l,o}}$, $C_{F3}^{l,o} = \alpha_3^{l,o} \times (\text{CAPE} \times \text{CLWC})^{\beta_3^{l,o}}$ and $C_{F4}^{l,o} = \alpha_4^{l,o} \times (\text{CAPE} \times \text{CSWC})^{\beta_4^{l,o}}$ where the parameters are listed in Table 1 for land, and Table 2 for ocean, respectively. These parameters are obtained from the best approximations of nighttime corona occurrence rate observed by ASIM as a function of values of CTH, CAPE $\times$ TP, CAPE $\times$ CLWC and CAPE $\times$ CSWC. The quality of the approximations covering the two years (1 April 2019 to 31 March 2021) is evaluated with $R^2$ metrics that results in $R^{2^l}$ and $R^{2^o}$ listed in Table 1 for land, and Table 2 for ocean, respectively. In all cases, the fitting produces a $R^{2^l}$ score equal or above 0.91 and a $R^{2^o}$ score equal or above 0.94, which indicate a strong correlation between BLUEs and the meteorological variables used. The dashed red (for land) and dashed blue (for ocean) lines in Figure 1(a)-(d) show the upper and lower limits of the fitting curve associated to the upper/lower errors in the fitting coefficients.

The parameterizations are implemented in EMAC using scaling factors as proposed in the lightning parameterization schemes (Tost et al., 2007). The applied scaling factors ensure a yearly occurrence rate of 3.5 BLUEs per second during the first year of present-day simulations. Also please note that the parameteriations are based on data of BLUEs, without including lightning data. Therefore, the developed corona schemes are independent of lightning. Regarding the sensitivity to corona discharges (BLUEs), we have calculated the spatial correlation coefficients (ranging between 0, no agreement, and 1, maximum agreeement) between the simulated and the observed climatology of BLUEs (see section 4.1).

## 4 Results

### 4.1 ERA5, CLARA and Model Simulations vs Observations

Figure 2 shows two-year average (1 April 2019 through 31 March 2021) nighttime geographical distribution of global corona (BLUE) electrical activity in thunderclouds according to the GD-2 distribution derived from ASIM observations (Soler et al., 2022) (a), and annual global predictions for BLUE occurrence rate based on hourly ERA5 data introduced in the corona (BLUE) parameterizations $C_{F2}$ (b), and $C_{F3}$ (c). Figure S1 in the supplementary material shows the BLUE occurrence rate based on hourly ERA5 data for $C_{F4}$. Note that the colorbars have the same scale. Considering that thunderstorms occur on hourly timescales, these plots show that when ERA5 hourly data are considered for the meteorological variables, the proposed BLUE

parameterizations are reasonably consistent with observations by ASIM. The global annual average BLUE rates obtained from the four adopted parameterizations using ERA5 hourly data (for CAPE, TP, CLWC and CSWC) and CLARA monthly data (for CTH) produce 3.50 BLUEs s$^{-1}$ for the four schemes.

Figure S2 and Figure S3 in the supplementary material show two-year average (1 April 2019 through 31 March 2021) nighttime geographical distribution of global corona (BLUE) electrical activity in thunderclouds according to the GD-2 distribution derived from ASIM observations (Soler et al., 2022) (a), and annual global predictions for BLUE occurrence rate based on CLARA monthly data introduced in the corona (BLUE) parameterizations $C_{F1}$, and monthly ERA5 data introduced in the corona (BLUE) parameterizations $C_{F2}$, $C_{F3}$, and $C_{F4}$.

Figure 3 illustrates the comparison between the two year average (1 April 2019 through 31 March 2021) nighttime corona GD-2 distribution observed by ASIM (a), and synthetic annual global average distributions obtained for the present day climate state model simulations applying the $C_{F1}$ (b), $C_{F2}$ (c), and $C_{F3}$ (d) cloud corona schemes, respectively. Figure S4 in the supplementary material shows synthetic annual global average distributions of cloud coronas obtained using $C_{F4}$. We calculated the synthetic annual global average by accounting for all time steps throughout the diurnal cycle assuming that daytime coronas in thunderclouds causing BLUEs are equally probable as those occurring at nighttime. Both, observations and simulation results, show four thundercloud corona chimneys clearly distinguishable over the Pacific ocean, the Americas, Africa / Europe and Asia / Australia. While the observed maximum in the Americas is located north of Colombia, model predictions locate it in a region partly covering the north of Peru, southern Colombia and eastern Brasil. Note that the spatial correlations (comparing observations in panel (a) with model simulations in panels (b)-(d) in Figure 3) are 0.4689 (for $C_{F2}$), 0.4542 (for $C_{F4}$), 0.4226 (for $C_{F1}$), and 0.3620 (for $C_{F3}$). Though the global annual average rate for observations and predictions is the same (3.5 coronas s$^{-1}$), the geographical distribution of coronas in thunderclouds predicted by the $C_{F2}$ scheme resembles ASIM observations more faithfully. For instance, according to available ASIM observations, the west of North America and the south of Australia hardly exhibit thundercloud corona activity. The Tornado Alley shows the highest activity in North America. These features are better reproduced by the $C_{F2}$ scheme than by the others (see Figure 4). In addition, the predicted cloud corona discharge geographical distribution in Africa / Europe given by the $C_{F2}$ scheme follows observations better than those predicted by the $C_{F3}$ scheme or the $C_{F4}$ scheme (shown in Figure S4 and Figure S5 in the supplementary material). Figure 4 shows the zonal (left panels) and meridional (right panels) nighttime geographical distributions of global corona (BLUEs) electrical activity in thunderclouds according to the GD-2 distribution derived from ASIM observations (orange line) (Soler et al., 2022), and zonal/meridional distributions of the annual global chemistry-climate model predictions (using 10 year simulations, blue line) for BLUE occurrence rate according to corona parameterizations $C_{F1}$ (top panel), $C_{F2}$ (middle panel), and according to $C_{F3}$ (bottom panel). The predicted zonal distributions show a reasonable agreement with observations. However, only the longitudinal distribution of $C_{F2}$ (middle panel) follows the observed longitudinal distribution in terms of the relative importance of the four BLUEs chimneys.

Thundercloud corona recordings by ASIM are conditioned by nighttime only observations, the South Atlantic Anomaly (SAA) and the relatively reduced number of detections due to the short period of observations (two years). These circumstances constrain the parameterizations developed in this study. However, when implemented in global atmospheric chemistry climate

models, simulations are able to predict BLUEs occurrence rate and geographical distribution in space-time regions where observations were not completely available. Regarding the SAA, please note that ASIM was not shut-off over the South Atlantic Anomaly (SAA), this was already discussed by Soler et al. (2022). Note that on March 2019 there was an update of the ASIM-MMIA cosmic ray rejection algorithm software (ON only over the SAA before March 2019, ON everywhere after March 2019) that could have influenced the originally obtained BLUEs global distribution, the so-called GD-1 (see Soler et al. (2021), and Figure 1 by Soler et al. (2022)). That was the main reason that moved us (in Soler et al. (2022)) to consider a new BLUEs dataset between 1 April 2019 and 31 March 2021, generating the so-called GD-2 Blue global distribution. The GD-2 distribution (used in this paper) already shown in Figure 2 of Soler et al. (2022) was generated by our modified BLUE search algorithm. This included a new condition (with respect to the algorithm originally presented by Soler et al. (2021)) consisting in that the 337 nm events (the associated 337 nm photometer light curve) are removed over the entire planet (not only in the SAA) when their rise times ($\tau_{rise}$) are $\leq 40$ $\mu$s and their total duration times ($\tau_{total}$) are $\leq 150$ $\mu$s (see Figure 2 by Soler et al. (2022)). Comparing Figure 1 (GD-1) and Figure 2 (GD-2, used in this paper) of Soler et al. (2022), it is clear that in GD-2 the Radiation Belt Particles (RBP) and Cosmic Rays (CR) are overall removed (not only in the SAA) but we think that, most probably, GD-2 underestimates the number of BLUEs. It should be clear that GD-2 is the ASIM observed distribution of BLUEs that we are adopting in this paper (see Figure 2(a)).

Figure 5 presents a comparison of the seasonal behavior of in-cloud coronas resulting from the GD-2 distribution of ASIM observations (left column) and the model predicted (right column) seasonal distribution according to the $C_{F2}$ corona scheme for the present day climate scenario. We notice that the $C_{F2}$ scheme correctly reproduces the seasonal global average occurrence rate, placing the maximum during the boreal summer (JJA) and the minimum in the boreal winter (DJF). In particular, ASIM nighttime observations indicate that the seasonal global average occurrence rates of in-cloud coronas are: 3.73 (SON), 2.60 (DJF), 3.75 (MAM) and 4.01 (JJA) events s$^{-1}$. Model predictions based on the $C_{F2}$ corona scheme result in 3.37 (SON), 3.26 (DJF), 3.39 (MAM) and 3.89 (JJA) events s$^{-1}$, respectively. The seasonal distributions of the $C_{F1}$, $C_{F3}$ , and $C_{F4}$ schemes are shown in the supplementary material (Figures S6, S7 and S8) and all of them exhibit their maxima in JJA (3.81 for $C_{F1}$, 3.96 for $C_{F3}$, and 3.93 events s$^{-1}$ for $C_{F4}$).

Model simulations based on the $C_{F2}$ scheme indicate that during the boreal winter (DJF), electrical activity in the form of thundercloud coronas is more important in the north of South America. However, this is not evident from DJF observations in South America, which exhibits disperse in-cloud corona activity probably due to limited ASIM observations in this region due to the presence of the SAA. The model predicted in-cloud corona discharge activity in the central and southern parts of Africa, its surrounding seas and the Asia / Australia region during the boreal winter is similar to available ASIM observations including the peak of thundercloud corona activity in the north of Australia.

The in-cloud corona observations by ASIM during the boreal summer season is in good agreement with model predictions. In particular, model simulations show that thundercloud corona discharges in Africa are mainly restricted to its central region, also significant in-cloud corona activity is predicted in India and surrounding seas including Indonesia and the eastern region of China. Model simulations also predict the largest number density of in-cloud coronas in North America during boreal summer, which is in agreement with ASIM seasonal observations of thundercloud corona discharges.

During the boreal autumn and spring seasons, the observed in-cloud corona maxima in America appear between the north of Colombia and central America reaching up to the west coast of Mexico (in the boreal spring). Model predictions of in-cloud activity maxima during these seasons point to southern Colombia and the maritime region next to the east coast of Mexico (in the boreal spring). ASIM observations indicate that corona activity in central Africa is more important during the boreal spring season than during the autumn, the same is predicted by the global chemistry-climate model using the $C_{F2}$ corona scheme.

Figure 6 and Figure 7 illustrate the end of the 21st century climate scenario and its comparison with present day. Figure 6 shows chemistry-climate simulations that result in the geographical distribution of global annual corona (BLUE) occurrence according to different BLUE parameterizations for the end of the 21st century (global annual average of years 2091 to 2095). Panels (a), (b), (c) and (d) correspond to end of 21st century BLUE occurrence rates of $4.02 \pm 0.04$ coronas s$^{-1}$ for $C_{F1}$, $4.04 \pm 0.07$ coronas s$^{-1}$ for $C_{F2}$, $5.33 \pm 0.03$ coronas s$^{-1}$ for $C_{F3}$, and $4.02 \pm 0.04$ coronas s$^{-1}$ for $C_{F4}$, respectively.

Figure 7 shows a comparison between predicted global annual average geographical distributions of in-cloud coronas under the present day conditions and by the end of the 21st century climate scenarios for the $C_{F1}$ (panel (a)), $C_{F2}$ (panel (b)), $C_{F3}$ (panel (c)), and $C_{F4}$ (panel (d)) thundercloud corona schemes. We find that, in general, the $C_{F3}$ corona scheme predict a large ($5.33$ coronas $\pm 0.03$ s$^{-1}$) global annual average occurrence rate for thundercloud coronas by the end of the 21st century when compared to present day simulations leading to $3.50 \pm 0.01$ events s$^{-1}$ (in agreement with ASIM nighttime observations). In

particular, for the end of the 21st century, the $C_{F2}$ and $C_{F3}$ based schemes predict global annual averages of 4.04 and 5.33 events s$^{-1}$, respectively. The standard deviation of the occurrence rates for thundercloud coronas in present day simulations range between 0.006 (for $C_{F1}$) and 0.008 for ($C_{F3}$), so that present day occurrence rates are significantly smaller than those of the end of the 21st century projections. Note that the p-values from a t-test comparing the periods 2001-2009 with 2091-2095 are $3.51 \times 10^{-12}$ (for $C_{F1}$), $2.41 \times 10^{-10}$ (for $C_{F2}$), $1.77 \times 10^{-19}$ (for $C_{F3}$) and $3.74 \times 10^{-13}$ (for $C_{F4}$).

As said above, one of the most noticeable features in the predicted RCP6.0 future scenario for the end of the century is the $5.33$ coronas $\pm 0.03$ s$^{-1}$ for the scheme $C_{F3}$. The underlying reason for this can be seen in the changes of the CLWC in Figure S9 (panel (c)) in the supplementary material. The liquid content at 440 hPa increases considerable over the continents because of the larger convection over the continents at the end of the 21st century. Moreover, there is less ice and snow at the 440 hPa pressure level so that Figure 8 shows that the scheme $C_{F4}$ produces less BLUEs in many continental regions.

Finally, the end of the 21st century climate scenario results in BLUE occurrence rates between 13 % larger ($\sim 3$ % per K) (for the $C_{F1}$, $C_{F2}$ and $C_{F4}$ schemes), and 52 % larger ($\sim 13$ % per K) (for the $C_{F3}$ scheme based on CAPE$\times$CLWC) than present day. According to chemistry-climate simulations (see Figure 6) for the end of the 21st century, the globally averaged temperature at the surface increases by about 4 K (Pérez-Invernón et al., 2023) by the end of the 21st century (RCP6.0 scenario, 2091-2095) compared with present-day scenario (2000-2009).

## 4.2   Robustness of BLUE parameterizations on the 1-hourly timescale

We have developed the four BLUE parameterizations using ERA5 hourly (for CAPE, TP, CLWC and CSWC) and COPERNICUS CLARA monthly (for CTH) data since the temporal resolution in chemistry transport models is of the order of minutes. To check that the BLUEs parameterizations behave reasonably well at model time scales, 1-hourly BLUE flash densities are

built for the period 1 April 2019 to 31 March 2021 for the 3 parameterizations ($C_{F2}$, $C_{F3}$ , and $C_{F4}$) as shown in Figure 8.

All the tested parameterizations exhibited approximately 95% of cells with values less than $10^{-4}$ Blues km$^{-2}$ h$^{-1}$. It can be seen that $C_{F4}$ produces the most homogeneous occurrence of BLUEs in time and space, while $C_{F2}$ exhibits the most inhomogeneous distribution, with more cells experiencing a high rate of BLUEs. The simulated distributions of BLUEs density in a chemistry-climate model can be compared with the distributions of flash density obtained by using lightning parameterizations (Finney et al., 2014, Fig. 5). The distributions of BLUEs obtained in this work show decreasing trends similar to those

of flashes concerning the number of cells versus the density value. Therefore, we can conclude that the parameterizations of BLUEs obtained in this study are as applicable to chemical-climate models as those of lightning.

## 5   Conclusions

Corona discharge activity in thunderclouds is found to be positively, but non-linearly correlated with: CTH, CAPE $\times$ TP, CAPE $\times$ CLWC, and CAPE $\times$ CSWC. These findings allowed us to develop four parameterizations for global chemistry-

climate model simulations of in-cloud corona activity distinguishing between land and ocean.

The four corona schemes were tested against a two-year dataset of worldwide nighttime in-cloud corona observations recorded by ASIM between 1 April 2019 and 31 March 2021. Projections to the end of the 21st century are also performed.

The global annual average BLUE rates obtained from the four adopted parameterizations using ERA5 annual averaged data (for CAPE, TP, CLWC and CSWC) and CLARA annual averaged data (for CTH) produce $\sim$ 3.50 BLUEs s$^{-1}$ for the four

schemes. The obtained spatial correlations are 0.4689 (for $C_{F2}$), 0.4542 (for $C_{F4}$), 0.4226 (for $C_{F1}$), and 0.3620 (for $C_{F3}$). Present day model predictions are in reasonable agreement with recent observations by ASIM.

Predictions for the end of the 21st century suggest BLUE occurrence rates range between 13 % larger ($\sim$ 3 % per K) (for the $C_{F1}$, $C_{F2}$ and $C_{F4}$ schemes), and 52 % larger ($\sim$ 13 % per K) (for the $C_{F3}$ scheme based on CAPE$\times$CLWC) than present day global average occurrence rate of BLUEs.

In-cloud corona schemes can be helpful to test global and / or regional chemical impact of corona discharges in thunderstorms since in-cloud coronas are known to directly produce not only greenhouse gases such ozone (O$_3$) and nitrous oxide (N$_2$O) but also oxidant species such as hydroxyl (OH) and hydroperoxyl (HO$_2$).

**Table 1.** Parameters for the thundercloud corona discharge parameterizations over land.

| Parameters | $C_{F1}^l$ | $C_{F2}^l$ | $C_{F3}^l$ | $C_{F4}^l$ |
|---|---|---|---|---|
| $\alpha_1^l$ | $0.046 \pm 0.017$ | | | |
| $\beta_1^l$ | $2.902 \pm 0.174$ | | | |
| $\alpha_2^l$ | | $12.711 \pm 0.832$ | | |
| $\beta_2^l$ | | $0.633 \pm 0.045$ | | |
| $\alpha_3^l$ | | | $523.096 \pm 135.673$ | |
| $\beta_3^l$ | | | $0.560 \pm 0.047$ | |
| $\alpha_4^l$ | | | | $451.034 \pm 100.006$ |
| $\beta_4^l$ | | | | $0.639 \pm 0.048$ |
| $R^{2^l}$ | 0.93 | 0.92 | 0.91 | 0.91 |

**Table 2.** Parameters for the thundercloud corona discharge parameterizations over ocean.

| Parameters | $C_{F1}^o$ | $C_{F2}^o$ | $C_{F3}^o$ | $C_{F4}^o$ |
|---|---|---|---|---|
| $\alpha_1^o$ | $0.006 \pm 0.002$ | | | |
| $\beta_1^o$ | $3.251 \pm 0.161$ | | | |
| $\alpha_2^o$ | | $2.199 \pm 0.106$ | | |
| $\beta_2^o$ | | $0.889 \pm 0.038$ | | |
| $\alpha_3^o$ | | | $341.741 \pm 83.971$ | |
| $\beta_3^o$ | | | $0.708 \pm 0.043$ | |
| $\alpha_4^o$ | | | | $129.313 \pm 22.393$ |
| $\beta_4^o$ | | | | $0.694 \pm 0.038$ |
| $R^{2o}$ | 0.94 | 0.98 | 0.96 | 0.96 |

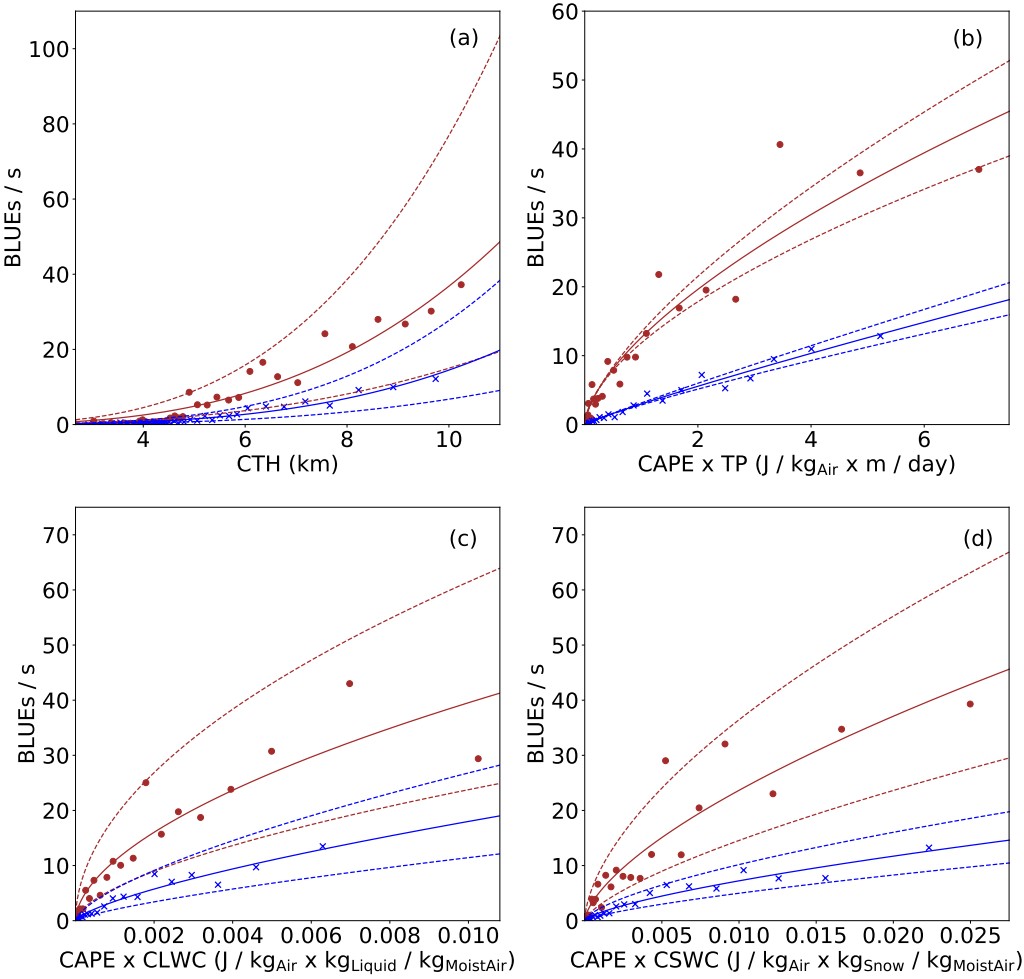

**Figure 1.** Global annual average occurrence rate (coronas s$^{-1}$) of nighttime thundercloud coronas observed by ASIM between 1 April 2019 and 31 March 2021 in land (red dots) and over the ocean (blue crosses) (a) versus corona discharge schemes based on the Cloud Top Heigh (CTH) (b), CAPE × TP (c), and versus the CAPE × specific CLWC (d), and versus CAPE × specific CSWC (d). The solid red (land) and blue (ocean) lines correspond to the functional forms for the nighttime corona occurrence rates of the four proposed schemes: $C_{F1}^{l,o} = \alpha_1^{l,o} \times (CTH)^{\beta_1^{l,o}}$, $C_{F2}^{l,o} = \alpha_2^{l,o} \times (CAPE \times TP)^{\beta_2^{l,o}}$, $C_{F3}^{l,o} = \alpha_3^{l,o} \times (CAPE \times CLWC)^{\beta_3^{l,o}}$ and $C_{F4}^{l,o} = \alpha_4^{l,o} \times (CAPE \times CSWC)^{\beta_4^{l,o}}$ where the fitting parameters (listed in section 3) are obtained from the best approximations of nighttime corona occurrence rate observed by ASIM over land and ocean as a function of values of CTH taken from the Essential Climate Variable (ECV) Cloud Properties of COPERNICUS (the European Union's Earth observation programme), from CAPE × TP, CAPE × CLWC and CAPE × CSWC taken from ERA5 reanalysis. The quality of the approximations covering the two years (1 April 2019 to 31 March 2021) is evaluated with $R^{2^{l,o}}$ metrics, which values are listed in section 3. The dashed red and blue lines in panels (a)-(d) show the upper and lower limits of the fitting curves associated to the upper/lower errors in the fitting coefficients over land and ocean, respectively.

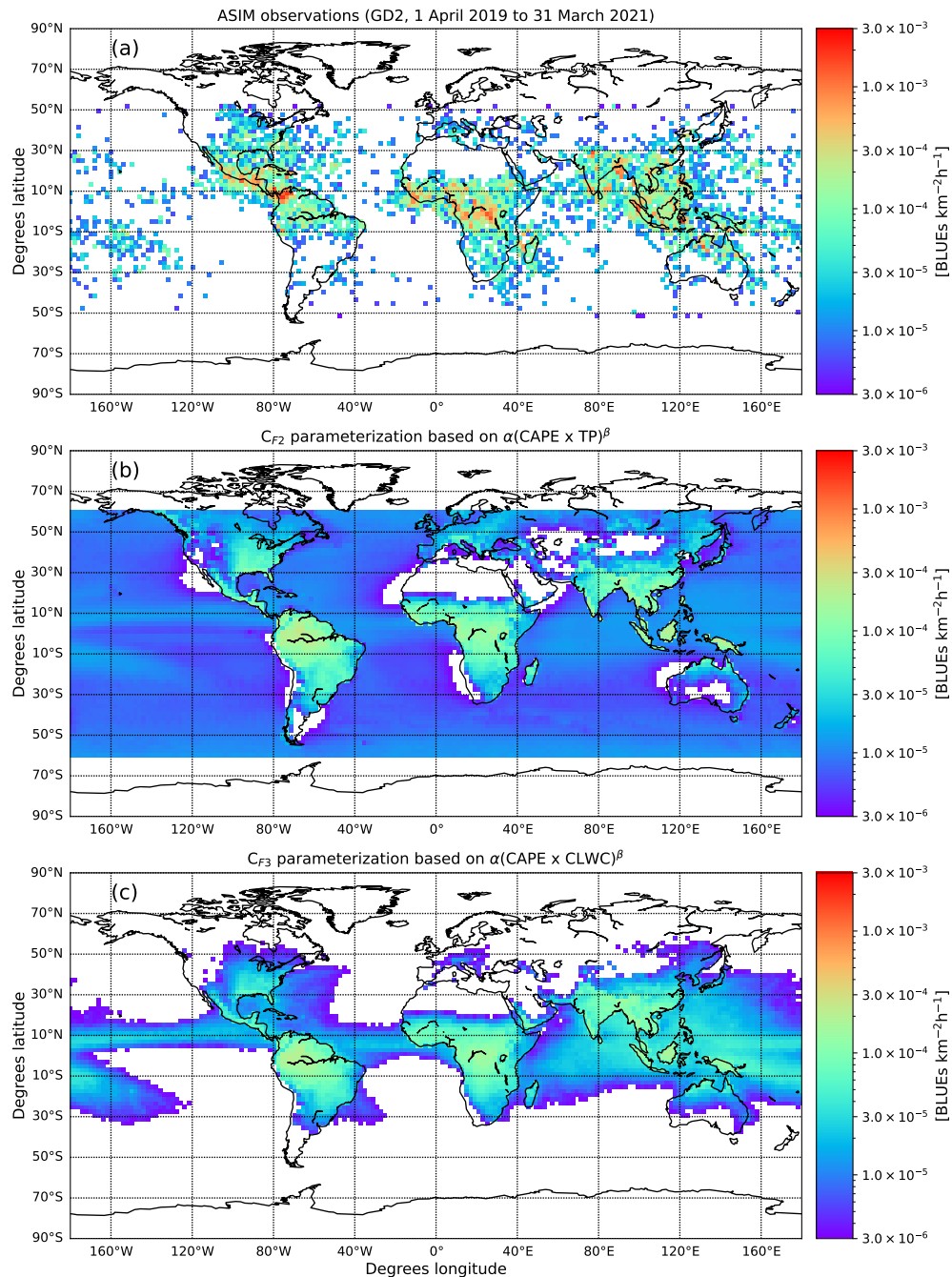

**Figure 2.** Two-year average (1 April 2019 through 31 March 2021) nighttime geographical distribution of global corona (BLUE) electrical activity in thunderclouds according to the GD-2 distribution derived from ASIM observations (Soler et al., 2022) (a), annual global predictions for BLUE occurrence rate based on hourly ERA5 data introduced in the corona (BLUE) parameterizations $C_{F2}$ (b), and $C_{F3}$ (c). Note that the colorbars have the same scale. Considering that thunderstorms occur on hourly timescales, these plots show that when hourly ERA5 data for the meteorological variables are considered, the proposed BLUE parameterizations work fine.

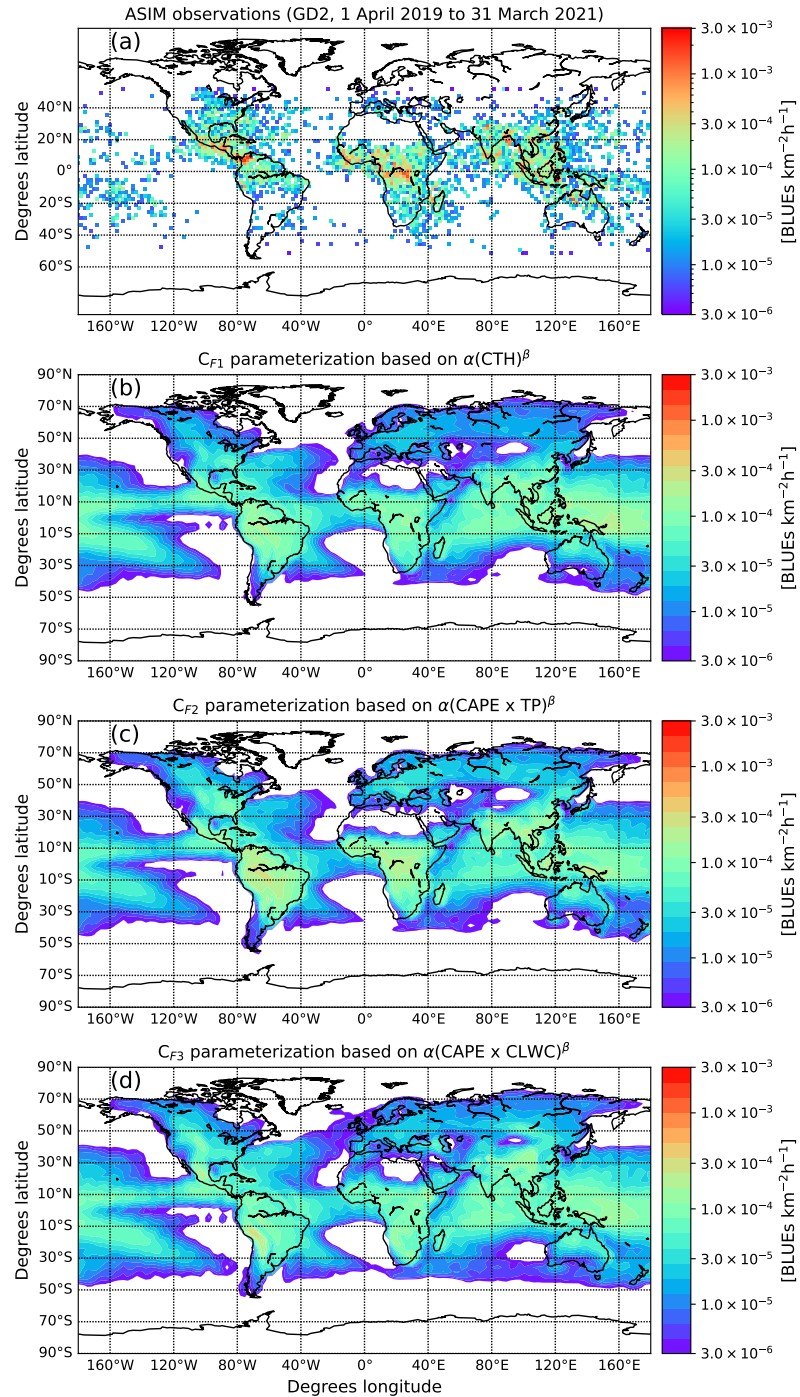

**Figure 3.** Two-year average (1 April 2019 through 31 March 2021) nighttime geographical distribution of global corona (BLUE) electrical activity in thunderclouds according to the GD-2 distribution derived from ASIM observations (Soler et al., 2022) (a), annual global chemistry-climate model predictions (using 10 year simulations) for BLUE occurrence rate according to corona parameterizations $C_{F1}$ (b), $C_{F2}$ (c), and according to $C_{F3}$ (d). Note that the colorbars have the same scale.

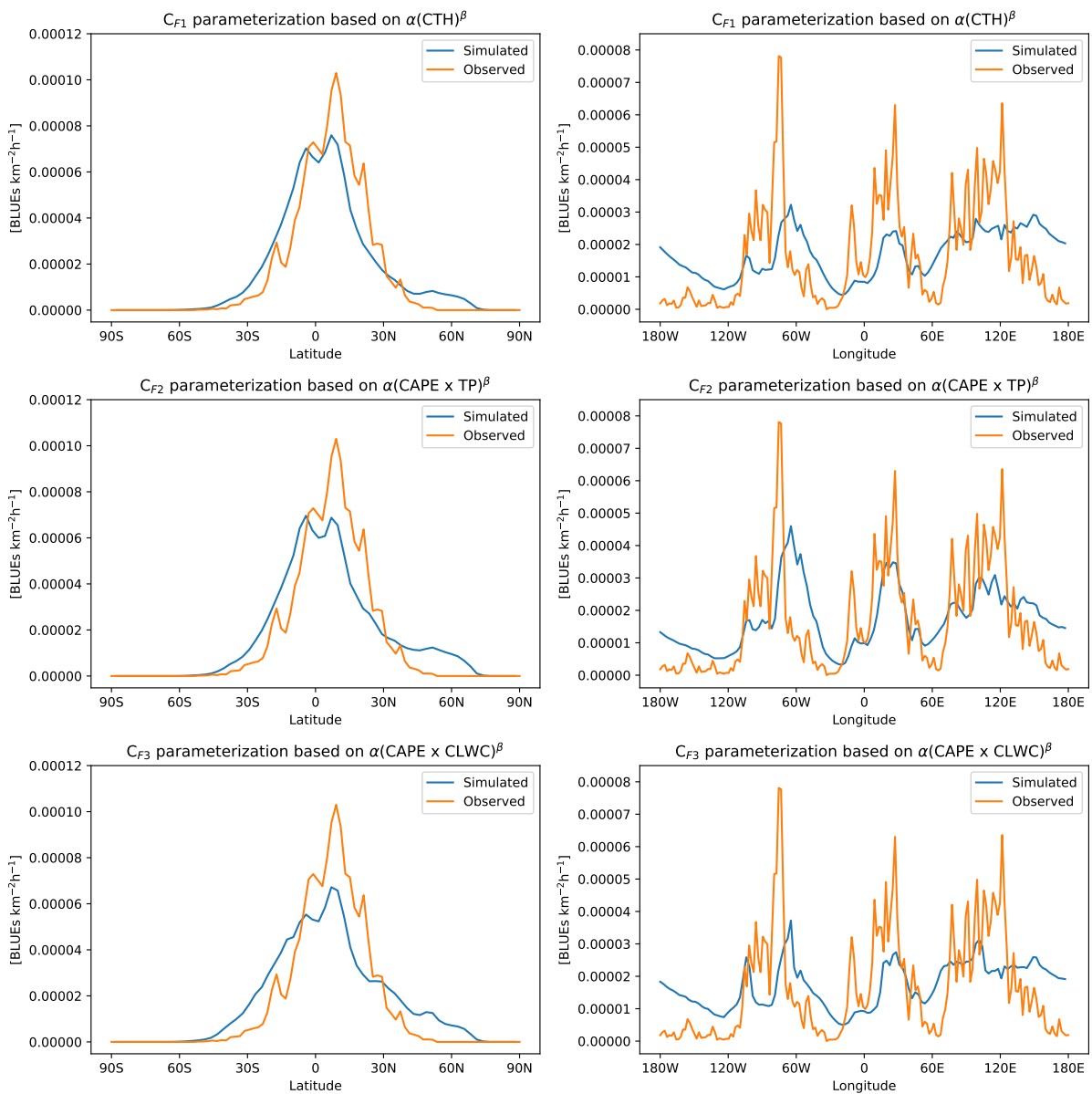

**Figure 4.** Zonal (latitudinal, left panels) and meridional (right panels) nighttime geographical distributions of global corona (BLUEs) electrical activity in thundercloud according to the GD-2 distribution derived from ASIM observations (orange line) (Soler et al., 2022), and zonal/meridional distributions of the annual global chemistry-climate model predictions (using 10 year simulations, blue line) for BLUE occurrence rate according to corona parameterizations $C_{F1}$ (top panel), $C_{F2}$ (middle panel), and according to $C_{F3}$ (bottom panel).

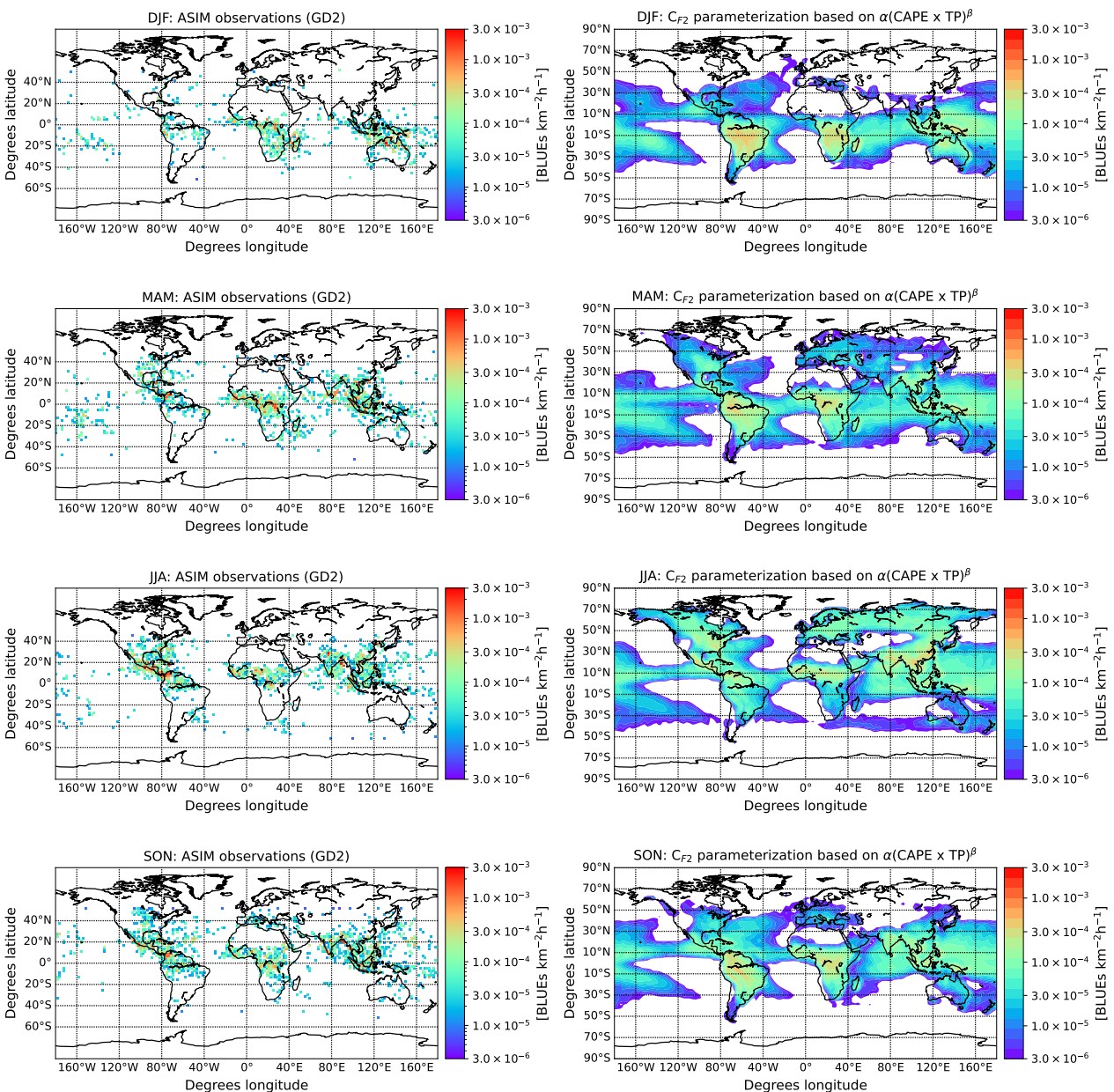

**Figure 5.** Two-year average (1 April 2019 through 31 March 2021) nighttime seasonal climatology of global corona (BLUE) electrical activity in thunderclouds according to GD-2 distribution derived from ASIM observations (Soler et al., 2022) resulting in 2.60 (DJF), 3.75 (MAM), 4.01 (JJA) and 3.73 (SON) coronas (or BLUEs) s$^{-1}$ (left column), and global annual average chemistry-climate model predictions (using 10 year simulations) for seasonal BLUE occurrence rate and geographical distribution according to the corona parameterization $C_{F2}$ resulting in 3.26 (DJF), 3.39 (MAM), 3.89 (JJA) and 3.37 (SON) coronas (or BLUEs) s$^{-1}$ (right column). Note that the colorbars have the same scale.

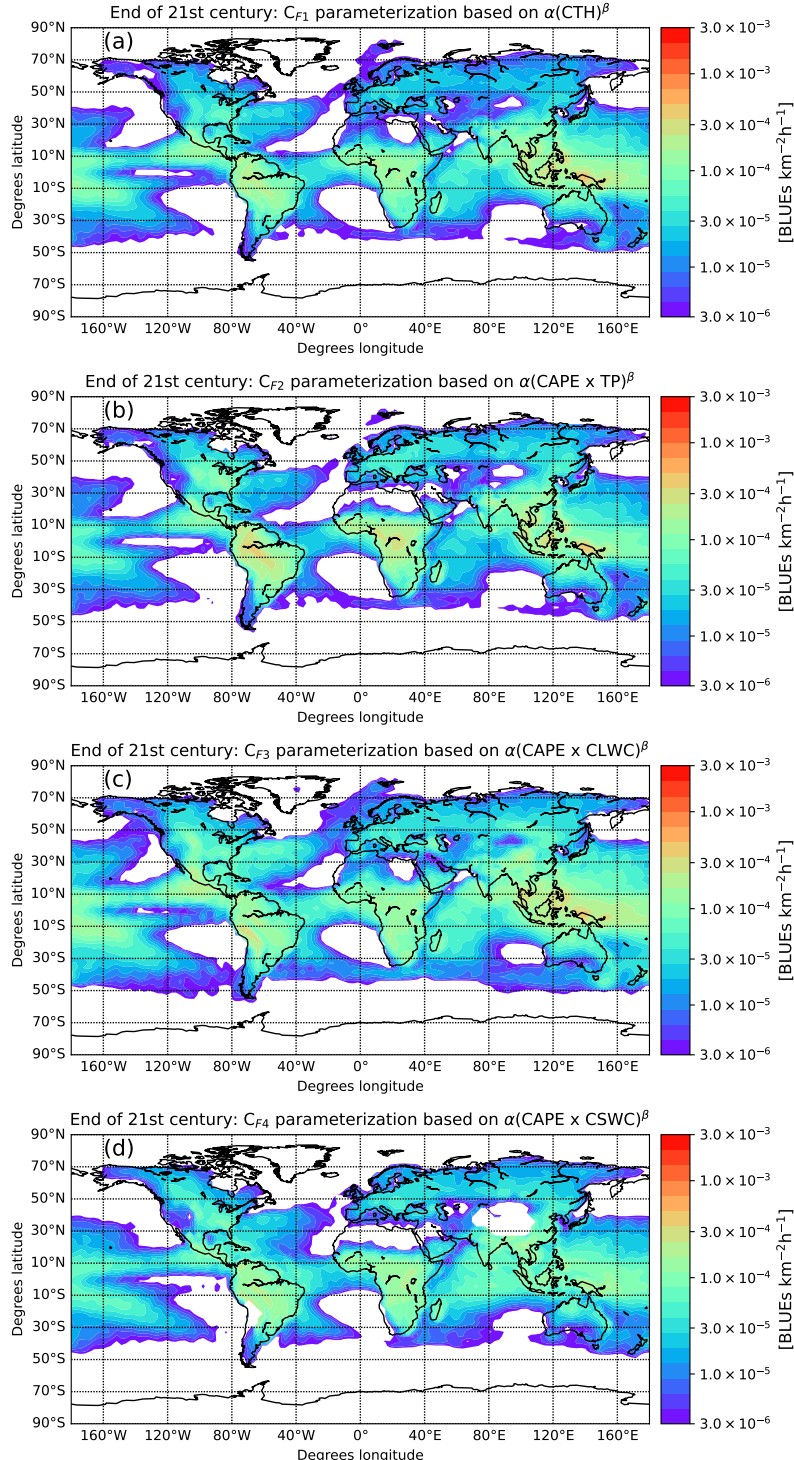

**Figure 6.** Chemistry-climate simulations showing the geographical distribution for the end of the 21st century (global annual average of years 2091 to 2095) of global annual corona (BLUE) occurrence according to different BLUE parameterizations. Panels (a), (b), (c) and (d) correspond to end of 21st century global annual average BLUE occurrence rate of 4.02 coronas $s^{-1}$, 4.04 coronas $s^{-1}$, 5.33 coronas $s^{-1}$ and 4.02 coronas $s^{-1}$, respectively. Note that the color-bars have been scaled.

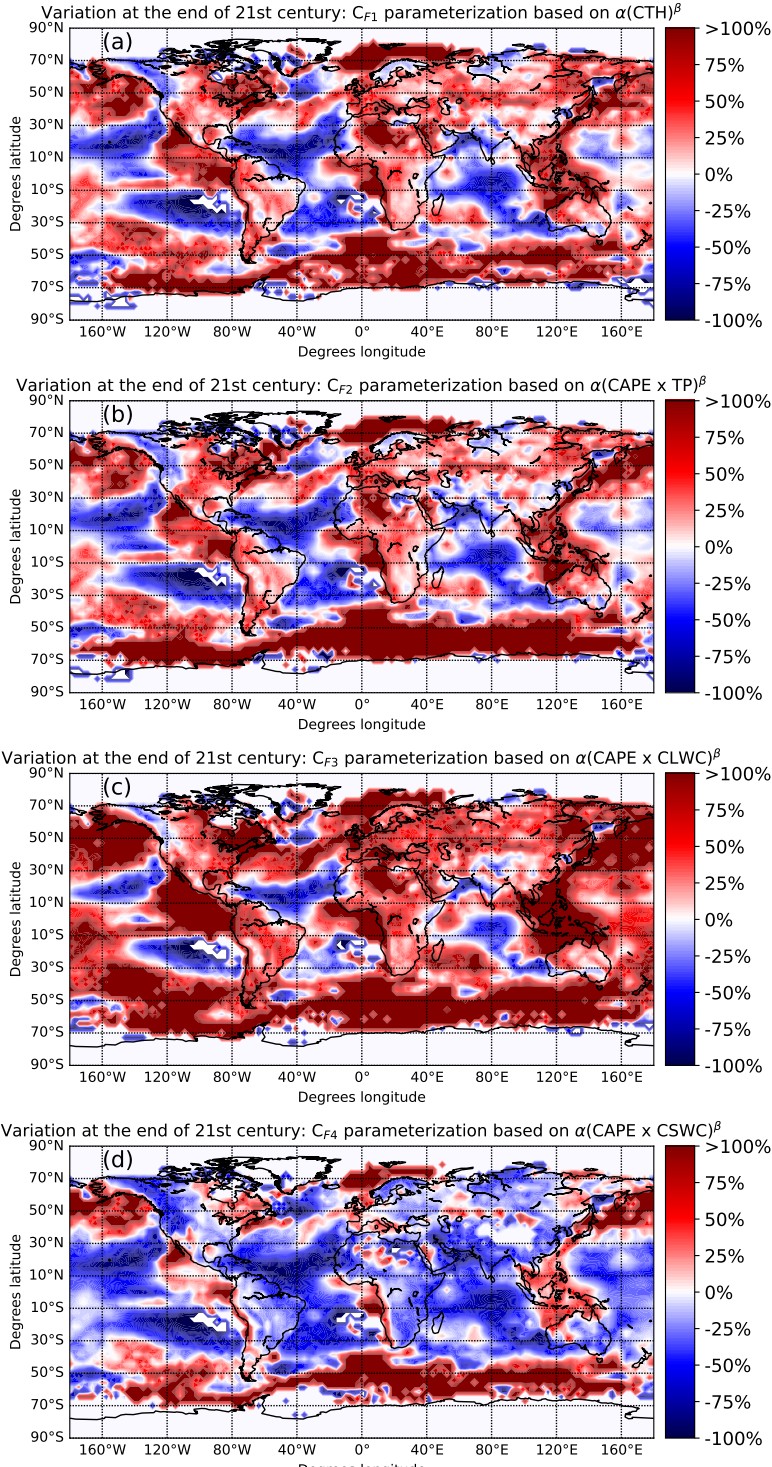

**Figure 7.** Chemistry-climate simulations showing the variation (percentage) between the geographical distribution for the end of the 21st century and that of present day (global annual average of years 2000 to 2009) of global annual corona (BLUE) occurrence according to different BLUE parameterizations. Note that the color-bars have been deliberately saturated at the upper ends due to the high variability of the plotted risk.

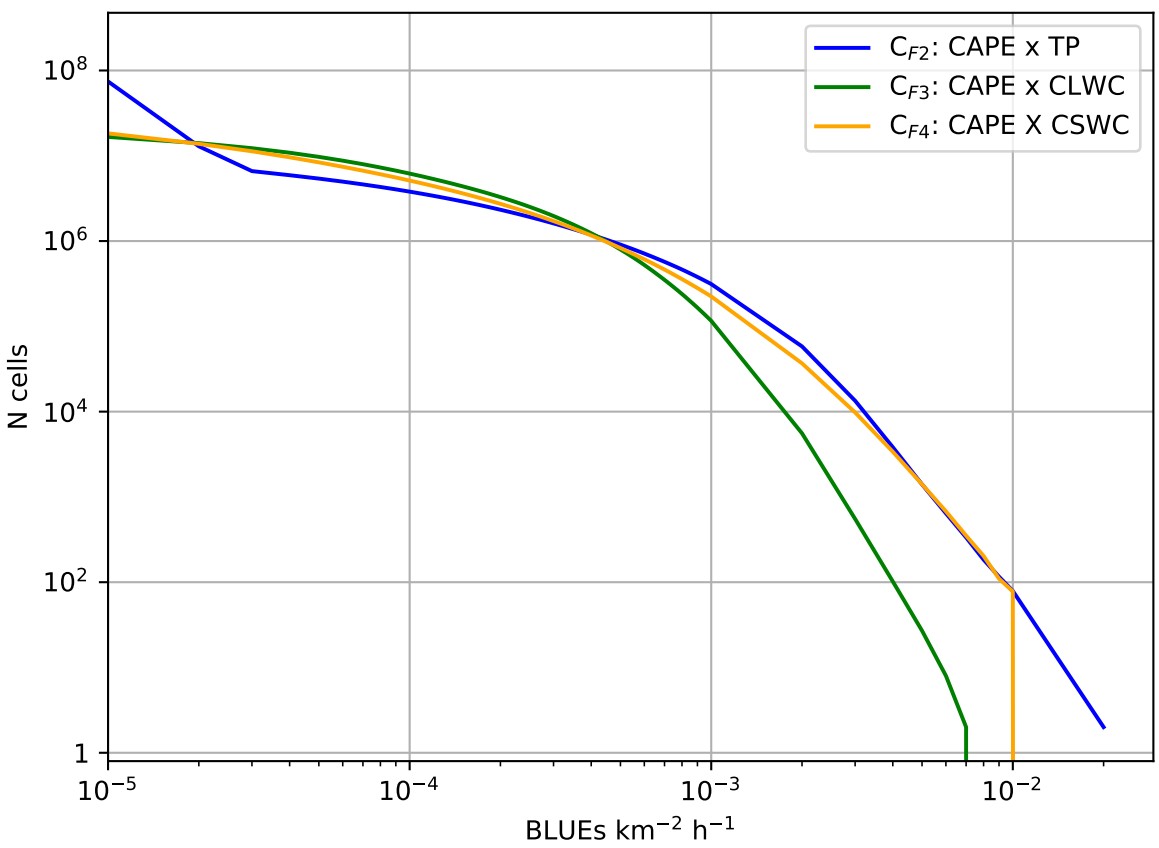

**Figure 8.** Curves show different 1-hourly BLUE flash densities in the period 1 April 2019 to 31 March 2021 for the $C_{F2}$ (blue line), $C_{F3}$ (green line), and $C_{F4}$ (orange line) BLUE parameterizations.

*Code and data availability.* The Modular Earth Submodel System (MESSy) is continuously developed and applied by a consortium of institutions. The usage of MESSy and access to the source code are licensed to all affiliates of institutions which are members of the MESSy

Consortium. Institutions can become a member of the MESSy Consortium by signing the MESSy Memorandum of Understanding. More information can be found on the MESSy Consortium website (http://www.messy-interface.org, last access: 22 November 2023). As the MESSy code is only available under license, the code cannot be made publicly available. The parameterization of in-cloud corona discharges has been developed based on MESSy version 2.55. ASIM level 1 data are proprietary and cannot be publicly released at this stage. Interested parties should direct their request to the ASIM Facility Science Team (FST). ASIM data request can be submitted through https://asdc.space.dtu.dk

(last access: 22 November 2023) by sending a message to the electronic address asdc@space.dtu.dk. The work performed was done by using the CLARA product data family (Karlsson et al., 2017) from EUMETSAT′s Satellite Application Facility (SAF) on Climate Monitoring (CM SAF): https://cds.climate.copernicus.eu/cdsapp#!/dataset/satellite-cloud-properties?tab=overview (last access: 22 November 2023). ERA5 reanalysis data Hersbach, H. et al. (2018a, b); Hersbach et al. (2020) were downloaded from the Copernicus Climate Change Service (C3S) Climate Data Store https://cds.climate.copernicus.eu/cdsapp#!/dataset/reanalysis-era5-single-levels?tab=overview (last access:

22 November 2023).The data from the simulations presented in this study are freely available under https://zenodo.org/records/12632821 (Soler et al., 2024)

*Author contributions.* S.S.: Methodology, validation, formal analysis, investigation. F.J.G.V. and F.J.P.I.: Conceptualization, methodology, validation, formal analysis, investigation, data curation, writing—original draft. P.J.: Methodology, validation, formal analysis, investigation, data curation, writing—review and editing. T.N., O.C., V.R. and N.O. Investigation, writing—review and editing.

*Competing interests.* At least one of the (co-)authors is a member of the editorial board of Atmospheric Chemistry and Physics.

*Acknowledgements.* This work was supported by the Spanish Ministry of Science and Innovation under projects PID2019-109269RB-C43, PID2022-136348NB-C31 and the FEDER program. SS acknowledges a PhD research contract through the project PID2019-109269RB-C43. FJPI acknowledges the sponsorship provided by Junta de Andalucía under grant number POSTDOC-21-0005, and by a fellowship from "La Caixa" Foundation (ID 100010434) with fellowship code LCF/BQ/PI22/11910026. Additionally, F.J.P.I. and F.J.G.V. acknowledge financial

support from the grant CEX2021-001131-S funded by MCIN/AEI/ 10.13039/501100011033. The high performance computing simulations (HPC) have been carried out in the DRAGO supercomputer of CSIC.

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
