# Peer review of "Parameterizations for global thundercloud corona discharge distributions"

_EGUsphere, 2024_

## Author Comment (AC1)

**Reviewer 2**

The manuscript is based on ASIM observation (BLUEs over cloud top) and make the parameter (CTH, CAPE, TP, CLWC, CSWC) fitting algorithm for BLUE occurrence rate using ECMWF ERA5 data. Then, authors used ASIM BLUEs occurrence rate to validate the adopted parameterization. Finally, they predict results with EMAC models and conclude that 17-28% large than present day model. The in-cloud corona schemes can help to understand the contribution of greenhouse gas and oxidant species from BLUEs.

I thoroughly enjoyed reviewing this manuscript and only have some minor requests for revision.

*Thank you very much for your constructive and encouraging comments that we appreciate. Please find below answers to your particular points.*

ASIM only recorded BLUEs at nighttime. Hence, the corona parameterizations with CAPE, TP, CLWC and CSWC were only validated at nighttime. In general, thunderstorm activity is expected to be more intense in the afternoon than nighttime since updraft are weaker without heating by sunlight. Are there any assumptions for BLUEs occurrence rate for nighttime or daytime?

*The referee is completely right in that thunderstorm activity is expected to be more intense in the daytime and afternoon than in nighttime. Please consider that we calculated the synthetic annual global average by accounting for all time steps throughout the diurnal cycle assuming that daytime coronas in thunderclouds causing BLUEs are equally probable as those occurring at nighttime (see section 4.1). In fact, we believe that our BLUE parameterization are completely valid for daytime (including afternoon) because during daytime, when updraft are stronger than during nighttime, the parameterization will use daytime values of the considered meteorological variables like CAPE, TP, CTH, CLWC and CLWC. However, since data of BLUE rates and geographical distribution are still missing (ASIM only measures during nighttime), we still cannot compare with BLUE daytime distribution. In the revised manuscript we have used the units of "Blues km-2 h-1" instead of "Blues km-2 night-1" or "Blues km-2 day-1" so that all figures become more consistent among them.*

The flash occurrence rate are several times larger than BLUEs. Is any significant difference between flash and BLUEs occurrence rate?

*This is an important question and a possible answer may come from looking at Fig 5b of a recent paper by Husbjerg et al. GRL 2022 (doi: e2022GL099 064), **cited in our manuscript**, showing that the CAPE associated to thunderstorms producing lightning flashes have a median value of 1000 J/Kg while thunderstorms producing BLUEs require stronger convection than needed from lightning alone. The CAPE found in the scenarios of thunderstorms that produce BLUEs range median values between 1280 J/Kg (slow BLUES, that is, those buried in the thunderclouds) and 1570 J/Kg (fast BLUES, that is, those appearing in the top of thunderclouds). As indicated in Husbjerg et al. GRL 2022, A CAPE greater than 2000 J/Kg usually indicates deep convection. Cells generating fast blue discharges have 25% occurrence in the region of deep convection. For cells generating only slow blue discharges it is 17% **while for regular lightning, only 10% of the events have a CAPE greater than 2000 J/Kg**. Therefore, there is a strong link between deep convection and the generation of blue discharge events. Another consequence is that it is then more probable that lightning occurs since they do not so much require the presence of deep convection to occur.*

Do you explain more about the contribution of greenhouse gas and oxidant species for BLUEs? Authors are encouraged to claim more important effects on the future weather system.

*Thanks for raising this point. However, that would be the subject of another paper in preparation so we prefer just to mention the possible influence of BLUEs in greenhouse and oxidant atmospheric gases.*

It is unclear that how the RCP6.0(Representative Concentration Pathway 6.0) affect the BLUEs occurrence rate? What is the important implication of climate changes for BLUEs rates?

*Figure S9 in the revised supplementary material shows the projected annual variation of the variables used to parameterize BLUEs under the RCP6.0 scenario. CAPE, total precipitation, and the cloud content of liquid and snow water are projected to increase in the regions (among others) with the higher occurrence of thunderstorms, such as Middle Africa, North America and Southeastern Asia. As a consequence, the global occurrence of BLUEs is projected to increase.*

Solar activity and aerosol from human activity may be related with climate change. In your modeling results, do you consider other external factors, e.g., solar radiation or aerosols and their relation to climate change. Bedsides, volcanic eruption or human activity will be the unexpected factors in your models.

*As detailed by Jockel et al. (2016), the future solar forcing has been prepared according to the solar forcing used for CMIP5 simulation of HadGEM2-ES, where the SSTs and SICs are taken from Jones et al. (2011; see also Sect. 3.3). It consists of repetitions of an idealized solar cycle connected to the observed time series in July 2008. Here, we deviate from the CCMI recommendations consisting of a sequence of the last four solar cycles (20–23).*

*Anthropogenic emissions are incorporated as prescribed emission fluxes following the CCMI recommendations (Eyring et al., 2013b). Troposheric and stratospheric aerosols are prescribed. In the case of RCP 6.0, anthropogenic emissions are taken from the RCP 6.0 data by Fujino et al., (2006). The anthropogenic emissions are prescribed from monthly values, which have been linearly interpolated from annual emission fluxes.*

*We acknowledge that volcanic eruptions and human activities are unexpected factors in the model. In fact, limiting our projections to the RCP 6.0 scenario is already a strong limitation, as other scenarios have been proposed, such as the RCP 2.6, RCP 4.5, RCP 8.5. More recently, the Shared Socioeconomic Pathways (SSP) SSP1, SSP2, SSP3, SSP4 and SSP5. However, we did not count with enough computational resources to simulate all the possible future scenarios.*

---

## Author Comment (AC2)

The manuscript by Soler et al. presents for the first time a series of parameterizations of thundercloud corona discharges, which have been extensively reported by the Atmosphere Space Interaction Monitor (ASIM) onboard the international space station in recent years. At a rate of about one tenth of that of lightning flashes, corona discharges greatly contribute to the electrical activity within thunderclouds and were predicted to have a non-negligible impact on the atmospheric chemistry. They may therefore be an additional and yet unknown source of several chemical compounds besides the upper tropospheric dominant lightning NOx source. The parameterizations are applied to reanalysis data and to a chemistry-climate model, allowing extension of the study to climate scenarios. I believe the manuscript is a novel and relevant contribution to the Journal, although I have some remarks and suggestions that I would like the authors to consider before publication. I apologize for the delayed publication of my comments.

*Thank you very much for your constructive and encouraging comments that we appreciate. Please find below answers to your particular points.*

My main concern is on the quality of the parameterizations, which are based on yearly and globally averaged data. Adopting yearly averaged data implies that we move away from the physical processes, which are linked to hourly or sub-hourly activity of the storms with a large temporal variability in a certain region (or grid point), towards the dependence on the geographical variability of average occurrences, which is then largely affected by large scale circulation and specific local conditions. I understand this is due to having only instantaneous observations at different spatial location, but the difference between temporal and geographical variability should be dealt with. The parameterization is in fact then applied to hourly data. The concern arises from the comparison to the observations, which seems not very satisfactory to me when looking at Fig. 2 (ERA5), Fig. 3 or Fig. 4 (model). At a first glance, the distributions found are very similar to climatologies of precipitation or lightning. The differences with the observations over the oceans and lightning chimneys are very large, roughly an order of magnitude. The parameterizations lead to almost homogenous peak values over the oceans and land in ERA5, missing the major lightning chimneys that are instead very clear in the observations. In the climate simulations, major peak values are shifted from the continents to the Pacific Ocean (160 E). I also miss a quantitative comparison of the spatial distributions. The quantitative agreement is purely based on global mean rates: the simulated rates are very good, but due to underlying large discrepancies in different regions. It seems that additional constrains over the ocean are needed, which in turn may lead to higher values over land. I think the highlighted discrepancies should be properly addressed.

*We understand and partially agree with the concerns of the referee. As the reviewer notes, please consider that ASIM is placed in the International Space Station (ISS), which rounds the Earth every 90 minutes approximately. Therefore, ASIM can only provide instantaneous observations in the different spatial positions of its low Earth orbit (LEO).*

*In order to improve our parameterizations, we have now considered additional constrains (as suggested by the referee) by distinguishing between ocean and land in each of the four schemes considered. The result is that, now, the parameterizations are not as homogeneous as before. They now clearly show the four lightning / BLUEs chimneys visible in the observational distribution of BLUEs. Also, we have included a new Figure (Figure 4) where we compare the observed (orange line) and simulated (blue line) latitudinal (zonal) and longitudinal (meridional) geographical*

*distributions of BLUEs. The comparison show that, now, the four chimneys are better captured by the simulations. The scheme based on CAPE and TP is the best since (a) it shows the best spatial correlation (0.4882) and (b) it keeps the same chimney relative importance as in the observations. Still, the Pacific ocean chimney is specially overrepresented in the simulations though the parameterization based on CAPE and TP shows the closest to the Pacific chimney observation.*

*Finally, as also suggested by the referee, we now provide a quantitative comparison of the spatial distributions that are now better than when no distinction was considered between land and ocean. The new (with land-ocean separation) spatial correlations (comparing observations with simulations) are 0.4882 (for scheme based on CAPE x TP), 0.4818 (for scheme based on CAPE x CSWC), 0.4540 (for scheme based on CTH), and 0.3910 (for scheme based on CAPE x CLWC). The previous spatial correlations with no land-ocean distinction were: 0.3510 (for the scheme based on CAPE x TP), 0.3354 (for scheme based on CAPE x CSWC), 0.3324 (for scheme based on CAPE x CLWC) and 0.2766 (for scheme based on CTH).*

A further concern I have is on the climate scenarios. They are interesting but I feel treated as a secondary product with no full support. As a result of this approach, they are also relegated to the supplementary only. I think either the authors are convinced by their results, and they should gain full presentation. Or they are not, and they should not be included. I think the projections are interesting and relevant, so they should find a proper description with at least one figure in the main paper. On the other hand, the changes that are presented are shown to be largely dependent on the land-ocean contrast, with large positive changes expected over the continents, and negative over the oceans. Since the adopted parameterizations fail to correctly simulate these contrasts, this will greatly affect future estimates. The projected changes may be different (much larger?) than what currently reported in the manuscript.

*We agree with the referee's concerns on this point. Now, in the revised version of the manuscript, we have adopted parameterizations for land and ocean and we think that the results improve and exhibit a better comparison with observations, including more faithful reproduction of the four lightning / BLUEs chimneys. This has consequences on the reliability of the simulations for future climate scenarios. Therefore, we are now confident in our future climate scenario simulations and have considered appropriate to show them in the main paper and not in the supplementary material.*

*Regarding the validity of the climate simulations (under the RCP6.0 scenario) please note that:*

*1.- We agree in that for the RCP6.0 simulations we consider only a few years (less than standard for being considered climate simulations). However, as mentioned in the text of the supplementary material, the calculated standard deviations (SD) of the frequency of BLUEs (in Blues per second) in present day are small (± 0.01) so that they do not overlap with those obtained for future (RCP6.0) climate scenarios.*

*2.- Although RCP6.0 simulations only cover 5 years, their initial conditions are those of 2090, i.e, the model is initialised with results from a previously performed transient climate simulation (for the same RCP6.0 scenario). Thus, the climate state of the end of the century in the model is fully established and the BLUEs parameterization, since based on meteorological parameters, adapts quasi immediately. This somehow means that it would be equivalent to start a simulation in 2010 until 2095 or to start it in 2090 since we are initializing the simulation of 2090 to the conditions existing in 2090 and not to those of 2010. For example, in 2090 we already have a global*

*temperature increase of about 4 K. We obtained that, as mentioned above, by initializing at 2090 the sea surface temperatures (SSTs), the sea-ice concentrations (SICs), the projected mixing ratios of the greenhouse gases and SF6, and the anthrophogenic emissions.*

*All the above lead us to think that it is justified to include climate simulations in the main paper.*

DETAILS

L39-45: this is hard to follow, please break sentence/rephrase.

*Done.*

L43: what is "6 – 3.5" standing for? If I read correctly, I would reverse that. Also, 45/6 and 45/3.5 does not give 7-12 times.

*The "6 – 3.5" stands for the maximum (6 Blues/s) at local midnight, and the average rate (3.5 Blues/s) of the nighttime global annual rate of Blues according to Figure 2 of Soler et al., 2022. However, the referee is right in that this was not clear enough. We have corrected it to avoid confusion.*

L47-48: I am not sure whether the results by Jenkins et al. 2021, and Brune et al. 2021, can be directly attributed to corona discharges. Could you elaborate better on this?

*We have used the word "could" intentionally so that the sentence is completely fair with the observations reported. Therefore, the suggested causality is degraded on purpose.*

*We cannot presently separate the corona and lightning stroke chemical contributions. So far it is still uncertain whether large streamer coronas underlying BLUEs and/or the numerous small coronas in lightning leaders are responsible of the sudden O3 increases. To be completely sure that the main cause is the occurrence of BLUEs, we would need to simultaneously measure the chemical and electrical activity in thunderclouds and conclude that **isolated BLUE activity** was dominant over lightning flashes at the time of the sudden enhancement of O3. To the best of our knowledge, those precise **simultaneous** observations have not yet been carried out. **Note that Brune et al., Science 2021, doi: 10.1126/science.abg0492 did not clarify what they meant by** "subvisible" discharges. However, **please note** that the recent laboratory experiments by Jenkins et al., JGR-Atm 2021, doi: 10.1029/2021JD034557 were designed to explain the aircraft DC3 observations (in summer 2012) reported by Brune et al., 2021. The lab experiments by Brune et al., 2021 clarify a bit the **meaning of the term "subvisible" (weak) electrical discharges**. According to **section 3.5 in the paper by Jenkins et al., 2021**, they specifically write (bold letters are ours):*

*"**Subvisible** electrical discharges expend much less charge than either sparks or corona. Since it was hard to produce subvisible discharges with the pointed electrodes, rounded electrodes with more uniform electric field gradients were used to obtain higher voltages before breakdown in order to examine the effects of subvisible discharges. Using the pulse width as the surrogate for energy, the pulse width was increased over the range from no electrical discharge, to subvisible discharge, to sparks. In Figure 5, LHOx, the OH exposure, LO3, the sum of LO3 and LNO2, and LNOx are shown for one of these experiments as a function of increasing discharge charge. The sum of LO3 and LNO2 is included as any NO2 likely originated from the reaction of O3 with NO, and therefore the total amount of O3 generated by the discharge would be the sum of LO3 and LNO2."*

*and section 5 continues with:*

*"At the lowest pulse widths, no visible discharge, corona, HOx, NOx, or O3 were detected. As the pulse width was increased, **increasing the energy discharged by the Tesla coil**, only OH and HO2 were detected. At greater pulse widths, O3 was measured. **Only when visible sparks were triggered at the electrodes was NOx detected**. **Prior to spark** onset, LHOx and the OH exposure had been steadily increasing with the pulse width, but both the LHOx mixing ratios and the OH exposure reached their maximum values at pulse widths near the transition to sparks, then decreased slightly for increasing pulse width after the transition to sparks. **LO3 increased slightly with pulse width up to the spark transition, then after the transition decreased as NOx converted most of the LO3 to LNO2**. LNOx continued to increase with increasing pulse width, consistent with the observations from the pointed electrodes (Figure S7)."*

*The above paragraph is somehow suggesting that at the spark (considered a reasonable laboratory "analog" of a lightning stroke) stage, O3 generation is stopped (in the lab). During the DC3 field campaigns described and analyzed by Brune et al., 2021, they detected O3 enhancements that they associated to "subvisible" discharges **because if lightning strokes** (and their leader coronas) were present, they would have seen them and, therefore, would not have used the term "subvisible" discharge. Thus, in our opinion, these two complementary studies (Brune et al 2021 and Jenkins et al 2021) suggest that dim (subvisible or hardly visible) streamer coronas (**not lightning leader coronas** inevitably accompanied by visible lightning leaders) in the observed and chemically probed thunderclouds described by Brune et al. 2021 could be responsible of those O3 enhancements.*

L50-59: this is a crucial point in your work but is not well introduced. It seems these meteorological parameters were used either because previously adopted or because they work pretty well. I would expect some minimal consideration on the physics behind. Also, it is not clear to the reader how this will be different from a lightning parameterization.

*Insights about the physical reasons underlying the selection of these corona parameterizations were already written in section 3 (between lines 151 and 169) of the manuscript. We have now added a reference to section 3 in lines 50-59 so that the reader can have a better idea about the reasons underlying the selection of the meteorological variables used.*

L58: please rephrase "seem to work pretty well"

*Done*

L60: here and elsewhere. Several sentences are very long and complex. Could the authors revise the manuscript breaking/shortening such sentences?

*Done*

L65: I understand the preliminary goal is to prove the adopted parameterizations work well. Please mention that the parametrizations are first tested on reanalysis data.

*Done*

L68: please note that 2091-2095 is not a climate relevant time interval. One should consider a 20 or 30-year long period if dealing with climate change.

*Please read our answer to the referee's second main point / concern above.*

L73: I find this unusual. If the authors explore the parameterizations in climate models under different climate scenarios, why relegating them to the supplementary only? If these are main results they should be in the main paper. If they are not, I would drop them.

*We have now moved simulations under different climate scenarios (Figure S6 in the supplementary material) to the main results. See our answer to the referee's second main concern above.*

L78-79: the sentence "ERA5 updates the previous ERA-Interim reanalysis (Dee et al., 2011) which were stopped being produced after 31 August 2019." is out of date. ERA5 has already been adopted by thousands of studies.

*Right. This is now changed accordingly.*

L86: this is unclear to me. Year data are produced and adopted for all parameters but (L91) the parameterizations are tested on hourly data. Could you clarify?

*ERA5 meteorological data can be downloaded on hourly or monthly temporal scales. It is important to differentiate between the temporal scale of the meteorological fields used to develop the parameterizations and the temporal scale of the meteorological fields used to test the parameterization:*

*1) Development of the parameterizations: The temporal scales of the climatology of BLUEs that we have developed are seasons and years (Soler et al. (2022)). Therefore, we cannot use hourly meteorological data to develop a parameterization of BLUEs. Hence, we use yearly meteorological data to develop the parameterizations.*

*2) Test of the parameterizations: Once the parameterizations are developed (Figure 1). We test their performance on hourly and monthly scales by using hourly and monthly meteorological data (Figure 2, S1, S2, S3).*

L99-100: Soler et al. 2022 is cited twice

*Corrected, thanks.*

L102: "candidates, this distribution is described" something wrong in the sentence here

*Correct sentence, improve it.*

L116: could you please specify what lightning parameterization is adopted in EMAC and whether the approach followed in this study is similar to Gordillo-Vázquez et al., 2019 by some of the same authors?

*The present day simulations without active chemistry are run without lightning chemical emissions. In the projection (climate) simulations, we have used the lightning parameterization by Grewe et al. (2001) based on the updraft flux of mass.*

L119-L120: are all these parameters obtained by subgrid parameterizations? Is this affecting your parameterization as compared e.g. to ERA5?

*Yes, these parameters are obtained by using subgrid parameterizations, which can affect the parameterizations of BLUEs. As in the case of lightning parameterizations, a scale factor is applied to the obtained BLUEs frequency to recover 3.5 BLUEs per second globally observed.*

*The parameterizations are implemented in EMAC using scaling factors as proposed for the lightning parameterization schemes (Tost et al., 2007). The applied scaling factors ensure a yearly occurrence rate of 3.5 BLUEs per second during the first year of present-day simulations.*

L131: please mention that the ERA-Interim starting field has no impact on the simulation (since you are then adopting ERA5 for the parameters).

*Done*

L138-143: I understand the limited period of the observations, which are of course due to the novel space experiment. It is on the contrary unclear to me why the authors have chosen such a limited period of time for the model simulation. I would expect some 20 years (or at least 10 years) to obtain enough variability under a climate scenario. Also, could the authors specify whether the EMAC model is run together with ECHAM, or whether the climate run was already available and the EMAC model is run starting from its results? This would clarify how a scenario run can be performed with only a 1 year spin-off.

*The present day simulations are run over 10 years. The projection simulations are run over 5 years. It is important to note that both simulations cover more than 2 years, which are the total number of years used to develop the parameterizations. We do not run more years because high computational resources are needed.*

*All the simulations in EMAC are run together ECHAM5.*

*The projection simulations are initialized by using the prescribed conditions of year 2090, previously obtained in the simulations RC2-base-04 of Jöckel et al. (2016). With this approach, the climate state of 2090 (as projected for the RCP6.0 scenario) is already established and the production of BLUEs, since based on the meteorological parameters, adapts quasi immediately.*

*Despite the projection simulation is run over 5 years instead of 10 years, the mean and the standard deviation of the obtained global rate of BLUEs is significantly larger than in present day simulations.*

L151-158: isn't CAPE simply showing the possibility of convection, rather than its actual occurrence? In fact, as you mention later, this is not working on the ocean. And moving to meridional distribution the correlation becomes fairly poor. It is not clear how often a high CAPE will be linked to lightning or BLUEs (see Husbjerg). Why not adopting the muCAPE?

***Figure 5b** of the paper by **Husbjerg et al. GRL 2022**, doi 10.1029/2022GL099064, showed that by clustering the BLUE discharge data set, cells which generate fast (close to cloud top) BLUE discharges have a median CAPE of 1390 $Jkg^{-1}$ compared to 1128 $Jkg^{-1}$ for cells generating only slow (deep in the cloud) BLUE discharges, further **indicating that stronger cells are more likely to generate fast BLUE discharge**. For comparison, the median CAPE for regular lightning was 816 $Jkg^{-1}$ in **Husbjerg et al. GRL 2022**, doi 10.1029/2022GL099064.*

*The above results lead us to use CAPE as a plausible meteo variable to track the occurrence of BLUEs. Thus, three of the proposed parametrizations are based on CAPE times another meteorological variable.*

L160: this is the first time slow and fast BLUE discharges are mentioned. Since this is a not well-known process, it would be of help for the reader knowing what slow and fast mean.

*Yes, we agree. In the revised manuscript we have briefly explained that the fast and slow terms underlie the scattering of the light emitted by BLUEs in thunderclouds. Since fast BLUEs are located in the cloud top, the scattering of their light emission is smaller (than that of slow BLUEs located in the bottom of the cloud). Consequently, the rise and decay times of the light curves (as seen by ASIM photometers) are faster than the rise/decay times of the slow BLUEs.*

L162: this is the first time that values for coronas are compared to values for lightning. I feel more relevance should be given since the beginning of the section (and of the paper) to this comparison.

*We have tried to somehow emphasize this point a bit more **in the introduction of the revised manuscript** so to, from that point on, readers are alerted that in-cloud coronas occur under not exactly the same meteorological conditions than lightning.*

How different do we expect the two distributions to be? How different are the driving processes/parameters?

*This is an important question and a possible answer may come from looking at Fig 5b of the paper by Husbjerg et al. GRL 2022 (doi: e2022GL099 064), **cited in our manuscript**, showing that the CAPE associated to thunderstorms producing lightning flashes have a median value of 1000 J/Kg while thunderstorms producing BLUEs **require stronger convection** than needed for lightning alone. The CAPE found in the scenarios of thunderstorms that produce BLUEs range **median values** between 1280 J/Kg (slow BLUES, that is, those buried in the thunderclouds) and 1570 J/Kg (fast BLUES, that is, those appearing in the top of thunderclouds). As indicated in Husbjerg et al. GRL 2022, A CAPE greater than 2000 J/Kg usually indicates deep convection. Cells generating fast blue discharges have 25% occurrence in the region of deep convection. For cells generating only slow blue discharges it is 17% **while for regular lightning, only 10% of the events have a CAPE greater than 2000 J/Kg**. Therefore, there is a strong link between deep convection and the generation of BLUE discharge events. Another consequence is that it is then more probable that lightning occurs since they do not so much require the presence of deep convection to occur.*

L165-166: as anticipated in the comments to the introduction, I feel there is too little explanation for adopting parameters that describe the liquid and frozen water content. This is the basis of the parameterization; I think the reader would much appreciate some better explanation.

*We have now added a discussion on the change of the meteorological fields between present day and projection simulations.*

*The liquid and the frozen water content are calculated by the CONVECT submodel of MESSy. See Jöckel et al. (2016) and Tost et al. (2006) for more details.*

*References:*

*Tost, H., Jöckel, P., and Lelieveld, J.: Influence of different convection parameterisations in a GCM, Atmos. Chem. Phys., 6, 5475–5493, doi:10.5194/acp-6-5475-2006, 2006b.*

L166: I do not understand why the authors cited He et al. Could you please clarify? Is it to support the use of CLWC and CSWC? But here the authors are not accounting for electrification processes. In fact, looking at the results over the ocean, this is possibly a source of the shortages that are shown. When are CLWC/CSWC transformed into charges? Can the authors impose here a correction to the discrepancies found particularly over the oceans?

*Yes, He et al (2022) is cited because we consider that the presence of water in the form of liquid and / or snow is a necessary (though not sufficient) condition for the occurrence of cloud electrification. Collision of graupel and ice water crystals at temperatures less than 253 K (-20 º C) results in a negative charge transfer to the graupel that falls to lower regions of the cloud. The lighter, positive charged ice crystals stay in the higher regions of the cloud.*

*The above clarifications are now mentioned in the introduction of the revised manuscript.*

L187: "predictand."?

*Done*

L195-205: the coefficients for the parametrizations are obtained fitting yearly global mean values. This leads to very large simplifications, which are then revealed by the spatial distributions in the following of the paper. Would the results be different if fitting directly convective precipitation or lightning? I.e., is the parametrization sensitive enough to corona discharges, or simply to lightning activity (or even convective precipitation)? If this is the case, what is the added value as compared to a lightning parametrization? This is not clear to me and I would very much appreciate seeing this discussed. This approach implies that only the average yearly conditions of a certain region are considered, rather than the full temporal variability within that region. Why not fitting the local (spatio-temporal) conditions and then average the results?

*The parameterizations are based on data of BLUEs, without including lightning data. The developed parameterizations are then independent on lightning.*

*Regarding the sensitivity to corona discharges, please note that we have now added the spatial correlation coefficient between the simulated and the observed climatology of BLUEs. We have then shown how the parameterization performs in a chemistry-climate global model.*

*Testing the parameterizations on a regional scale could be achieved by implementing the parameterizations on regional models. However, this is is out of the scope of this paper.*

L210: I think the manuscript should make the different use of hourly, monthly and yearly data clearer. If I understand correctly, here the parameterizations are applied to ERA5 hourly data and then averaged over the time interval. This was not completely clear to me since the parameterizations are derived from annual mean data.

*This has been addressed before.*

L213: "the proposed BLUE parameterizations work fine and are consistent with observations by ASIM". I think this is overstated. Looking at Figure 2, I can see the parameterization leads to roughly 1 order of magnitude differences over the main lightning chimneys (too little) and over the ocean (too much). Not a word is currently spent on the discrepancies over the oceans.

*The sentence highlighted by the referee is now downtoned. Also, please note that in the revised manuscript each parameterization distinguishes between land and ocean. We think the agreement over the main lightning chimneys has now improved (specially in the case of the scheme based on the product of CAPE and TP), including the previous discrepancies over the oceans.*

The authors should compare the observations and parameterized occurrence density quantitatively (difference, ratio, R^2, RMSE, etc). Also, adopting two different style of contour/bin mapping does not help, and I invite the authors to adopt a consistent approach to ease the comparison. One way could be downgrading the spatial resolution (e.g. 5x5 or even 10x10) and showing both observations and results with the same mapping style. I understand some interesting features at the edge of the active regions would be smeared out, but the comparison would be more robust. The same concern applies to the results from the climate model in Figure 3. Here the deficiencies over the ocean are even larger, with values in the western Pacific exceeding those over Africa. Once the comparison is improved, I feel these shortages should be tackled somehow, imposing further

constrains over the oceans. Right now, the very good agreement on global rates depend on a counterbalance between large discrepancies.

*As suggested by the referee, we now provide a quantitative comparison of the spatial distributions that are now better than when no distinction was considered between land and ocean. The new **(with land-ocean separation)** spatial correlations (comparing observations with simulations) are 0.4689 (for scheme based on CAPE x TP), 0.4542 (for scheme based on CAPE x CSWC), 0.4226 (for scheme based on CTH), and 0.3620 (for scheme based on CAPE x CLWC). The spatial correlations **with no land-ocean distinction** were: 0.3510 (for scheme based on CAPE x TP), 0.3354 (for scheme based on CAPE x CSWC), 0.3324 (for scheme based on CAPE x CLWC) and 0.2766 (for scheme based on CTH). The climate projections are now also run considering parameterizations that distinguish land and ocean.*

L257-258: the authors should better describe this: is it ASIM shut-off over the SSA or not? How are the observations affected by the SSA?

*ASIM was not shut-off over the South Atlantic Anomaly (SAA), this was already discussed in Soler et al., 2022. Note that, as discussed in Soler et al., 2022, on March 2019 there was an update of the ASIM-MMIA cosmic ray rejection algorithm software (ON only over the SAA before March 2019, ON everywhere after March 2019) that could have influenced the originally obtained BLUEs global distribution, the so-called GD-1 (see Soler et al, 2021, and Figure 1 in Soler et al., 2022). That was the main reason that in Soler et al. 2022 moved us to consider **a new BLUEs dataset between 1 April 2019 and 31 March 2021** (and not between 1 September 2018 and 31 August 2020 as in Soler et al., 2021), generating the so-called GD-2 Blue global distribution. The GD-2 distribution (used in this paper) already shown in Figure 2 of Soler et al., 2022 was generated by our modified BLUE search algorithm that included a new condition (with respect to the algorithm originally presented in Soler et al., 2021) consisting in that the 337 nm events (the associated 337 nm photometer light curve) are removed in the entire planet (not only in the SAA) when their rise times ($\tau_{rise}$) are ≤ 40 μs and their total duration ($\tau_{total}$) times are ≤ 150 μs (see Figure 2 in Soler et al. 2022). Comparing Figure 1 (GD-1) and Figure 2 (GD-2, used in this paper) of Soler et al., 2022, it is clear that in GD-2 the Radiation Belt Particles (RBP) and Cosmic Rays (CR) are removed in all the planet (not only in the SAA) but we think that, most probably, GD-2 underesttimates the number of BLUEs. It should be clear that GD-2 is the ASIM observed distribution of BLUEs that we are adopting in this paper (see Figure 2(a)).*

L271: Why relegating climate projections to the supplementary. If these are robust, I think they deserve to have at least one figure in the text. On the other hand, since the agreement over the ocean is poor, also the estimates in projections will be affected. In particular, most of the negative changes are shown to occur over the ocean.

*The referee is right. In the revised manuscript we have moved the revised Figure S6 to the main manuscript after considering land-ocean in each parameterization. In the new Figure S6 we see that ...*

Have the authors obtained their changes (e.g., 28% or 24%... depending on the type of parameterization) by a global average of the changes? Or as a change in average global rates in the past and in the future? I think the latter case would be more robust, since regions with little contribution will continue to have little contribution even if increasing by a large amount. The discrepancies over the ocean will greatly affect these estimates since projections show large

negative differences over the ocean only. One may therefore expect a much larger positive change on a global average than currently estimated in the manuscript.

*We have now added a figure showing the differences at every grid cell and a figure showing the present day and the projected spatial distributions of BLUEs (new Figure 6 in the revised manuscript). We hope that the spatial variations are now clearer.*

*In order to obtain the changes (28%, 24%, etc…) we have calculated the total number of BLUEs per year from present day simulations and from projection simulations. The regional changes are now shown in the new figure 7 of the revised manuscript.*